

# The Open Global Glacier Data Assimilation Framework (AGILE) v0.1

Patrick Schmitt[1], Fabien Maussion[1,2], Daniel N. Goldberg[3], and Philipp Gregor[4]

[1]Department of Atmospheric and Cryospheric Sciences, University of Innsbruck, Innsbruck, Austria
[2]School of Geographical Sciences, University of Bristol, Bristol, United Kingdom
[3]School of Geosciences, University of Edinburgh, Edinburgh, United Kingdom
[4]Meteorologisches Institut, Ludwig-Maximilians-Universität München, München, Germany

**Correspondence:** Patrick Schmitt (patrick.schmitt@uibk.ac.at)

**Abstract.** The growing availability of glacier observations poses a challenge for models to integrate this heterogeneous information in a dynamically consistent way. At the same time, estimates of current glacier volume and area remain uncertain, as many global inventories and thickness datasets date back to the early 2000s. We present the Open Global Glacier Data Assimilation Framework (AGILE), a time-dependent variational method inspired by 4D-Var data assimilation. AGILE is built on a reimplementation of the OGGM flowline glacier evolution model in PyTorch, enabling full differentiability through automatic differentiation (AD). We test AGILE v0.1 in a series of idealized experiments designed to reflect common initialization and calibration scenarios in global glacier modeling. The goal is to recover glacier bed topography and distributed ice volume in 2020 through transient calibration, based on dynamical simulations starting in 1980. In these experiments, we assume a perfectly known mass balance and fixed ice dynamics parameters. While this setup simplifies real-world complexity, it allows us to isolate and evaluate the core functionality of the approach. Our results show that AGILE efficiently optimizes multiple control variables by leveraging AD-derived gradients, requiring only a few iterations to substantially improve upon initial guesses. We also examine the potential to reconstruct earlier glacier states (e.g., in 1980) without direct observations and find that this is fundamentally limited by the diffusive nature of glacier dynamics, even in an idealized setting. Overall, our experiments demonstrate AGILE's potential as a flexible and efficient data assimilation framework. Its ability to integrate diverse datasets in a dynamically consistent manner makes it a promising tool for future real-world glacier modeling applications.

## 1 Introduction

Mountain glaciers are retreating globally (The GlaMBIE Team, 2025; Hugonnet et al., 2021; Zemp et al., 2015), causing significant impacts on sea level rise (e.g., Marzeion et al., 2020; Rounce et al., 2023; Slangen et al., 2022; Zemp et al., 2019), freshwater resources (e.g., Aguayo et al., 2024; Huss and Hock, 2018; Ultee et al., 2022; Wimberly et al., 2024), and ecosystems (e.g., Bosson et al., 2023; Cannone et al., 2008; Gobbi et al., 2021). Predicting these impacts requires dynamic glacier evolution models that rely on past observations to simulate glacier behaviour into the future. However, many glaciers are located in remote and hard-to-access regions, making direct measurements difficult to obtain and scarce. As a result, global models are highly dependent on Earth observation (EO) data from satellites.



Early EO datasets mainly included glacier outlines (e.g., RGI 6. Consortium, 2017; RGI 7 Consortium, 2023) and digital
elevation models (e.g., COP-DEM, 2022; NASA JPL, 2020). Recently, more glacier-specific datasets have become available
(e.g., Hugonnet et al., 2021; Millan et al., 2022), allowing global glacier models to shift towards using glacier-specific obser-
vations for calibration rather than relying on regional averages or data from a few well-studied glaciers (Zekollari et al., 2024;
Marzeion et al., 2012). This shift is reflected in the design choices of the eight large-scale glacier models participating in the
recent Glacier Model Intercomparison Project 3 (GlacierMIP3; Zekollari et al., 2025).

The GlacierMIP3 models adopt different calibration and initialization strategies, but share many common features and rely
on the same datasets. The initial glacier surface geometry and corresponding start date are defined using RGI version 6 outlines
(RGI 6. Consortium, 2017) and their associated timestamps. Mass balance is calibrated separately by matching to a geodetic
mass balance observation over a period of approximately 20 years, using data from Hugonnet et al. (2021) (seven models) or
Shean et al. (2020) (one model), while assuming a fixed glacier geometry during this period. All models rely on the consensus
ice thickness estimates from Farinotti et al. (2019), either by directly using the distributed fields or by matching total glacier
volumes on a glacier-specific or regional basis.

Three models include a spin-up during initialization to account for glacier evolution prior to the outline date. In these
cases, a past glacier state is defined, and the model is run forward in time to match specific targets. The Community Ice
Sheet Model v2.1 (CISM2; Lipscomb et al., 2019) matches the ice thickness field, the Global Glacier Evolution Model ice flow
(GloGEMflow; Zekollari et al., 2019) targets the glacier-specific total volume and glacier length from the outline, and the Open
Global Glacier Model (OGGM; Maussion et al., 2019) uses the glacier total area, total volume and geodetic mass balance to
additionally refine its previously calibrated mass balance by accounting for evolving surface geometry (Aguayo et al., 2024).

Outside GlacierMIP3, an adapted version of OGGM's dynamic spin-up was extended to incorporate a second glacier outline
in a small region of the Alps, enabling the model to simultaneously match observed area and volume changes at regional scale
for the first time (Hartl et al., 2024). This enhancement allowed performance improvements when tested against additional
validation data and increased confidence in the model's projections through 2100 compared to the setup relying solely on
globally available datasets.

Another recent approach in the Alps is presented by Cook et al. (2023), who use the Instructed Glacier Model (IGM; Jouvet,
2022), a deep learning based 3D ice flow model. IGM assimilates various observations such as distributed thickness data
(GlaThiDa; Welty et al., 2020) and surface velocity fields (Millan et al., 2022) to initialize glaciers in a dynamically consistent
state. The transient simulation then begins from this point, removing the need for a spin up. However, all inputs must correspond
to the same timestamp, such as the outline date, since the method is limited to snapshot inversions. IGM also focuses only on
ice dynamics and bed reconstruction and does not include a dedicated mass balance model.

Despite these efforts, current approaches face several limitations. Most models neglect past glacier evolution during initial-
ization and do not account for surface geometry changes during calibration. They are tailored to currently available global
datasets and struggle to incorporate new observation types or repeated measurements in a seamless way, limiting their flex-
ibility to use all available data on a glacier-specific scale. It is also often implicitly assumed that all observations apply at
the outline timestamp. Furthermore, many methods rely on computationally intensive optimization schemes where only one





parameter is adjusted at a time, or they apply steady state assumptions to manage the general problem of overparameterization
in global glacier modelling (e.g., Rounce et al., 2020).

To overcome these limitations, an ideal calibration method should ensure dynamic consistency, be able to use temporally
distributed observations, avoid assumptions about the glacier's dynamic state, and allow the simultaneous optimization of
multiple model parameters. It must also remain computationally efficient to be applicable at regional and global scales.

To move closer to the transient calibration of large scale glacier models, we present a proof-of-concept for the Open Global
Glacier Data Assimilation Framework (AGILE). AGILE is based on a time-dependent variational assimilation approach, in-
spired by 4DVar methods (Lorenc, 1997). It iteratively adjusts all control variables of a dynamic glacier evolution model during
a transient (forward) simulation to minimize a cost function that quantifies the mismatch between model output and available
observations. To ensure computational efficiency, AGILE is implemented in PyTorch, a machine learning framework (Paszke
et al., 2019), which enables the use of automatic differentiation (AD). AD-based methods have previously been applied to snap-
shot inversions in regional glacier modeling (Cook et al., 2023) and in ice sheet modeling (Brinkerhoff and Johnson, 2013), as
well as to transient assimilation in ice sheet modeling (Goldberg and Heimbach, 2013; Recinos et al., 2023). To our knowledge,
AGILE represents the first application of a transient AD-based assimilation approach in the context of global glacier modeling.

In this study, we introduce AGILE v0.1, which reimplements the dynamic flowline model of OGGM in PyTorch to make it
fully differentiable through AD (Sect. 2). We evaluate the implementation using idealized experiments with synthetic glaciers
and observations designed to reflect globally available datasets (Sect. 3). The objective is to initialize a dynamically consistent
glacier by optimizing bed topography and ice volume distribution as control variables, assuming a perfectly known mass
balance and fixed dynamic parameters. While these assumptions simplify real-world complexity, they allow us to isolate and
assess how effectively AD guides the optimization of control variables during a transient model run (Sect. 4). We conclude by
discussing the broader implications of our results and outlining future directions for real-world applications (Sect. 5).

## 2  Methods


AGILE is developed as an open-source extension of the Open Global Glacier Model (OGGM) framework (Maussion et al.,
2019), with the code publicly available at github.com/OGGM/AGILE, and version 0.1 archived on Zenodo (Schmitt et al.,
2025). The following sections provide a detailed description of the underlying methodology and its individual components.

### 2.1  General principles

The general approach of AGILE is illustrated in Figure 1. A set of initial control variables, referred to as the first guess
and which may represent model parameters, boundary conditions, or both, is iteratively adjusted to minimize the mismatch
between the output of a glacier evolution model and available observations. This mismatch is quantified by a cost function,
which combines the discrepancies across all observational targets into a single value, with each contribution normalized by its
associated uncertainty. In addition, regularization terms are included in the cost function to address the ill-posed nature of the





problem and to reduce the risk of overfitting. The optimization aims to minimize this cost function by systematically adjusting the control variables.

To support this process efficiently, AGILE leverages automatic differentiation (AD) through PyTorch. This enables flexible definition of control variables while still allowing the calculation of their gradients with respect to the cost function. These gradients guide the optimization and allow for efficient and simultaneous updates of all variables. More technical details about
the AD implementation in PyTorch are provided in Appendix A1.

## 2.2 Glacier representation and forward model

AGILE re-implements the glacier evolution model from OGGM (Maussion et al., 2019) in PyTorch, making it fully differentiable. The model is based on the shallow ice approximation to compute depth-integrated ice velocity (equation A3) and a mass conservation equation, where changes in ice thickness at a point equal the mass balance input minus the divergence of
the ice flux (equation A4). Since version 1.6.0, OGGM uses a semi-implicit numerical scheme instead of the scheme described in Maussion et al. (2019). Details on this scheme are provided in Appendix A2.

AGILE uses the same 1.5D flowline representation as OGGM. In this approach, glacier dynamics are computed along a flowline with variable surface widths. The flowlines are derived from geographical input data, including glacier outlines and a digital elevation model (DEM), using the 'elevation band flowlines' method (e.g. Huss and Farinotti, 2012; Huss and Hock,
2015; Werder et al., 2019), which preserves the glacier's area-height distribution. Each grid point along the flowline is assigned a trapezoidal bed shape with a 45° wall-slope, and the bottom width depends on the glacier bed height (see Sect. 2.3.1 for more details). The 1.5D flowline setup reduces spatial complexity compared to full 2D or 3D models, making it computationally feasible to track the full computational graph during model runs to be used for AD, without running into memory limitations (see Appendix A1 for details about AD in PyTorch).

To integrate height-dependent Mass Balance (MB) forcing into the computation of gradients, AGILE includes a PyTorch wrapper (see Appendix A3 for details). This accounts for changes in MB forcing caused by dynamically evolving glacier surface heights. The wrapper allows the use of any OGGM-compatible MB model for dynamic simulations without re-implementing it in PyTorch. However, this approach does not support gradient computation for MB model parameters (e.g., the melt factor in temperature index models), but this can be added in the future.

## 2.3 Cost function

The cost function is defined as

$$\mathcal{J}(\mathbf{\Theta}) = \mathcal{J}_{obs}(\mathbf{\Theta}) + \lambda \mathcal{J}_{reg}(\mathbf{\Theta}), \tag{1}$$

where $\mathcal{J}obs$ represents the target observational component (see Sect. 2.3.1), and $\mathcal{J}reg$ is the regularization component (see Sect. 2.3.2). The control variables are denoted with $\mathbf{\Theta}$ (see Sect. 2.4), and $\lambda$ determines the weight of the regularization term
relative to the observational term.




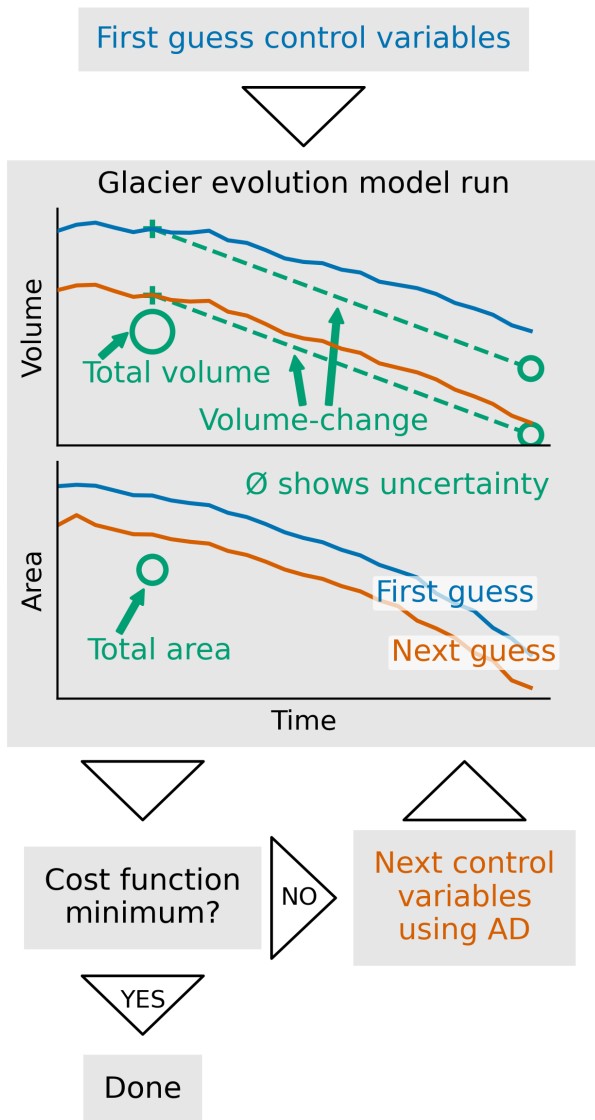

**Figure 1.** Principle workflow of AGILE: Starting with a First Guess at the top, an initial glacier evolution model run is performed (blue line). The mismatch between the model output and observations, in particular one total volume, one volume change and one total area, is calculated using a cost function. If the cost function has not reached a minimum, all control variables are updated simultaneously using AD, and a new glacier evolution model run is conducted (orange line).





### 2.3.1 Target observational cost component

The target observational component, $\mathcal{J}_{obs}$, measures the alignment of the model with the target observations using squared differences. For a single observed quantity $x$, it is defined as

$$\mathcal{J}_{obs,x} = \frac{(x_{mdl} - x_{obs})^2}{\sigma_x^2},$$ (2)

where $x_{mdl}$ is the modeled quantity, $x_{obs}$ is the corresponding target observation, and $\sigma_x$ represents the uncertainty of the target observation. This ensures that discrepancies within the uncertainty bounds have values smaller than one and also scales different types of target observations. For multiple observations, $\mathcal{J}_{obs}$ is averaged over all normalized discrepancies.

$$\mathcal{J}_{obs} = \frac{1}{n_{obs}} \sum_{j}^{n_{obs}} \mathcal{J}_{obs,j},$$ (3)

where $n_{obs}$ equals the number of considered target observations.

In this study, we focus on target observations that are readily available from global datasets. Specifically, a single area–height distribution, one geodetic mass balance value ($\Delta M$), and one total glacier volume estimate ($V$). The choice of these variables and the implications for real-world applications are discussed in Sect. 1.

The area–height distribution is derived from a glacier outline and a DEM, and is also used to define the elevation-band flowlines (see Sect. 2.2). As a result, we aim to match the surface heights ($sfc$) at each grid point along the flowline at the
outline year. By doing so, we also match the area–height distribution, since it was used to construct the flowlines. To maintain this relationship during minimisation, the bottom width of the trapezoidal bed shape is adjusted at each iteration based on the current bed height estimate (see control variables in Sect. 2.4). This preserves the link between surface height and the corresponding elevation-band area at each grid point.

As an example, the target observational cost for volume is given by

$$\mathcal{J}_{obs,V} = \frac{(V_{mdl} - V_{obs})^2}{\sigma_V^2},$$ (4)

while for surface height along the flowline it is

$$\mathcal{J}_{obs,s} = \frac{\sum_{i=1}^{n_x}(sfc_{mdl}^i - sfc_{obs}^i)^2}{\sigma_{sfc}^2 n_x},$$ (5)

where, $n_x$ is the number of grid points and $sfc^i$ is the surface height at grid point $i$. Here, the volume represents a glacier-integrated observation, while the surface heights are an observation along the flowline, incorporating the observed area-height
distribution. AGILE's flexible cost function design allows for expanding the range of supported target observations as long as corresponding model counterparts exist.





### 2.3.2 Regularization cost component

While $\mathcal{J}obs$ ensures alignment with target observations, the problem can remain ill-posed or prone to overfitting to observations. Regularization, $\mathcal{J}_{reg}$, introduces constraints to address these issues, typically focusing on smoothness (e.g. Goldberg and Heimbach, 2013; Fürst et al., 2017; Jouvet, 2022). The regularization term is scaled to ensure compatibility with $\mathcal{J}_{obs}$, similar to the observation uncertainties.

In this study, we apply regularization solely to enforce a smooth glacier bed. This is defined as

$$\mathcal{J}_{reg,bed} = \frac{1}{\gamma_{bed}} \sum_{i=0}^{n_x-1} \left( \frac{b^{i+1} - b^i}{\Delta x} \right)^2, \tag{6}$$

where $b^i$ is the glacier bed height at grid point $i$, $\Delta x$ is the grid spacing, and $\gamma_{bed}$ is the scaling factor based on the smoothness of the first guess bed height $b_{fg}$ ($\gamma_{bed} = \sum_{i=0}^{n_x-1} \left( \frac{b_{fg}^{i+1} - b_{fg}^i}{\Delta x} \right)^2$). The summation in this equation starts at $i = 0$, because at the start of the flowline (the highest point), an additional grid point is introduced to preserve the bed slope. Without this extra grid point, the algorithm would lower the bed height at the highest point to match the height of the second-highest point, minimizing the smoothness term. This effect gets stronger with larger values of $\lambda$. The extra grid point acts as an additional regularization to counteract this effect, regardless of the value of $\lambda$.

For our experiments we defined $\mathcal{J}_{reg} = \mathcal{J}_{reg,bed}$. However, this regularization framework could be extended in future studies, for instance, to include smoothness constraints on the initial flux. Such extensions will become increasingly important when working with real-world target observations, which are subject to measurement uncertainties. In these cases, regularization plays a crucial role in preventing the model from overfitting to noisy data.

### 2.4 Control variables

Control variables, denoted as $\boldsymbol{\Theta} = [\Theta_1, \Theta_2, ..., \Theta_n]$, are adjusted at each iteration to minimize the cost function (see Sect. 2.3). They represent the unknown parameters or variables to be estimated or optimised. In theory, these can include any unknown aspect of the system, such as flowline-geometry parameters (e.g. glacier bed height), dynamic parameters (e.g. the deformation-sliding parameter A of Equation A3), MB model parameters (e.g. melt factor), or a combination of these. The flexibility of AD enables the simultaneous calculation of gradients for multiple control variables, providing essential information to guide the minimization algorithm (see Sect. 2.5). However, this is constrained by observational data availability and the risk of overfitting.

In this study, we set the control variables to be the glacier bed elevation and the distributed ice volume in 1980 at each grid point. This choice reflects our primary goal: to demonstrate, as a proof-of-concept, that variables with different types, meanings, and magnitudes can be optimized simultaneously within each iteration. We also aim to explore whether AGILE can (i) improve upon the current OGGM bed inversion, which assumes equilibrium and may introduce biases for non-steady-state glaciers, and (ii) reconstruct glacier conditions in 1980. The second objective is of secondary importance and mainly serves to enable transient simulations from 1980 to 2020. This allows us to incorporate target observations at their respective timestamps,





such as the area-hight distribution and the total volume both in 2000, as well as the geodetic mass balance from 2000 to 2020. The goal is to obtain a dynamically consistent glacier state in 2020, which is not part of the optimization but is important for 180 initializing future projections.

To handle control variables with varying magnitudes, we apply min-max normalization. For a given control variable $\Theta_x$, this scaling is defined as

$$\Theta_{x,scaled} = \frac{\Theta_x - \Theta_{x,min}}{\Theta_{x,max} - \Theta_{x,min}}, \qquad (7)$$

where $\Theta_{x,min}$ and $\Theta_{x,max}$ represent the defined minimum and maximum bounds, respectively. This ensures all scaled variables 185 are within the range [0, 1], making changes proposed by the minimization algorithm approximately proportional across all variables. The minimum and maximum values for the bed heights are defined such that the resulting ice thickness at the time of the observed outline and DEM remains within $\pm60\%$ of the first-guess estimate. Similarly, the 1980 ice volume at each grid point is constrained within $\pm40\%$ of the first guess. Additionally, ten extra grid points are added beyond the terminus for the ice volume, allowing the glacier to be initialized with a larger extent than observed. For these extra points, the scaling boundaries 190 are set to 0 and the maximum value at the terminus grid point (140% of the first guess volume).

We also must account for the non-uniform grid used in the 1.5D flowline representation (see Sect. 2.2), where grid-cell areas vary along the flowline. These variations influence glacier-wide properties when control variables are changed. For example, a change in the bed height affects the glacier area-height distribution more in a larger grid cell than in a smaller one. To account for this, we multiply the glacier bed height by the initial surface width at each grid point and use the resulting values as our 195 control variables. The initial surface width is set during flowline initialization to preserve the observed area-height distribution and remains constant throughout minimization (see Sect. 2.2). We use width instead of grid-cell area because the spacing along the flowline is the same for all grid cells, making it a constant factor. The 1980 glacier volume at each grid point naturally accounts for varying grid-cell sizes. For the actual control variables, we divide the volume by the constant grid-cell spacing along the flowline and use the resulting volume per unit length as the control. This corresponds to the cross section area at each 200 grid point.

## 2.5 First guess and minimization

To initiate the algorithm, an initial estimate of the control variables, specifically the glacier bed height and the 1980 volume distribution, is required. In our experiments, we used two methods to generate these estimates, allowing us to assess AGILE's sensitivity to the first guess. The first method relies on the default OGGM bed inversion, a mass-conserving approach based 205 on an equilibrium assumption and an apparent mass balance, as described in Maussion et al. (2019). The second uses the shear-stress-based GlabTop method (Linsbauer et al., 2012), which depends solely on surface geometry.

In both cases, the estimated bed geometry directly provides the initial values for both the bed height and the 1980 volume distribution. However, it is important to note that the volume estimates from both methods correspond to the outline year (2000), and we introduce a temporal mismatch by using them as the first guess 1980 volume distribution.





210    The minimization process is conducted using the L-BFGS-B algorithm (Byrd et al., 1995; Zhu et al., 1997; Morales and Nocedal, 2011), as implemented in the Python package SciPy (Virtanen et al., 2020, https://www.scipy.org/). This algorithm, commonly used in similar applications (e.g., Goldberg and Heimbach, 2013; Fürst et al., 2017), supports bounded constraints for the control variables. We set the bounds for the control variables to our defined minimum and maximum values as described in Sect. 2.4.

## 3    Experiments

The goal of these experiments is to demonstrate that AGILE can recover both the glacier bed heights along the flowline and the dynamic glacier state in 2020 by performing a transient simulation from 1980 to 2020. A key objective is to show that AGILE improves upon OGGM's bed inversion, which assumes glacier equilibrium and may introduce errors for retreating or advancing glaciers, while at the same time producing a dynamically consistent glacier state in 2020 that could serve as the initial condition for future projections. To achieve this, we define the control variables as the glacier bed height and the 1980 ice volume at each grid point (see Sect. 2.4).

The target observations are based on globally available datasets, used here in an idealized setting. Specifically, we use an area–height distribution and a total volume estimate for the year 2000, along with a geodetic mass balance between 2000 and 2020. This setup is considered idealized because we exclude uncertainties in both mass balance forcing and ice dynamics. While this simplification does not reflect real-world complexity, it serves as a proof of concept, demonstrating that AGILE is correctly implemented and that gradients obtained with AD can be used to optimize multiple control variables simultaneously.

To create a controlled test environment, we generate synthetic glaciers using OGGM and treat the model output as observations. This gives us complete knowledge of the system, allowing us to directly assess the accuracy of our methodology. This type of setup, often referred to as an "inverse crime" (Colton and Kress, 2013), offers an optimistically favourable testing environment. However, demonstrating robust and reliable performance under such idealized conditions is an essential first step for validating a new inversion method (Goldberg and Heimbach, 2013).

### 3.1    Creation of synthetic glaciers and measurements

The idealized experiments are designed to replicate realistic glacier geometries. We selected the glaciers Aletsch, Artesonraju, Baltoro, and Peyto because they represent a range of different climates and glacier area size. For each glacier, a flowline is constructed using the RGI v6 outline (RGI 6. Consortium, 2017), DEM data from NASADEM (NASA JPL, 2020), and the consensus ice thickness estimate (Farinotti et al., 2019). Based on these flowlines, we generate dynamic glacier simulations from 1980 to 2020 for three different dynamic states: retreating, advancing, and equilibrium.

In particular, the retreating and advancing cases allow us to evaluate whether AGILE can improve upon OGGM's bed inversion, which assumes a glacier is in equilibrium. The resulting glacier geometries, shown in Figure 2 panels a and b for Aletsch, and in Figure S1 panels a and b for Artesonraju, panels e and f for Baltoro, and panels i and j for Peyto, serve as the ground truth we aim to invert for.





The driving mass balance is defined using a simple degree-day model (see Equation 1 in Schuster et al., 2023), with precipitation and temperature inputs taken from the W5E5 dataset (Lange, 2019) for each glacier's location. Precipitation factors (ranging from 1.4 to 5.2), degree-day factors (3.1 to 6.5 kg m$^{-2}$ °C$^{-1}$ day$^{-1}$), and temperature biases (-5.6 to 0.9 °C) follow the calibration values from OGGM v1.6. The deformation-sliding parameter $A$ (ranging from 0.1 to 7.9 * 10$^{-24}$ s$^{-1}$ Pa$^{-3}$) is also taken from OGGM v1.6, leading to different dynamic behaviors for each synthetic glacier.

An additional temperature bias is applied on top to create either retreating or advancing glaciers. Equilibrium glaciers are generated using a mean mass balance profile calculated over a specific time period from the W5E5 data. Figure 2 panels c and d show the resulting mass balance and volume evolution for the different dynamic states of the Aletsch glacier. For the other glaciers, see Figure S1 panels c and d for Artesonraju, panels g and h for Baltoro, and panels k and l for Peyto.

During the creation of the synthetic glaciers, we "measure" the target observations used in our experiments. Specifically, we record the surface elevation at each grid point in the year 2000 ($sfc_{2000}$) for capturing the area height distribution (for details see Sect. 2.3.1), the total glacier volume in 2000 ($V_{2000}$), and the geodetic mass balance from 2000 to 2020 ($\Delta M_{2000/2020}$). We define the associated measurement uncertainties as $\sigma_{sfc} = 10$ m, $\sigma_{\Delta M} = 100$ kg m$^{-2}$ yr$^{-1}$ and $\sigma_V = 10\%$ of $V_{2000}$. These target observations are designed to reflect the type of readily available global datasets and represent the only inputs provided to both the first-guess methods and AGILE in our experiments.

### 3.2 Performance measurements

To evaluate how well AGILE can reconstruct the glacier bed and the dynamic glacier state in 2020, we track the mean absolute difference (MAD) of key variables throughout the optimization process. Specifically, we compute the MAD of the glacier bed elevation (MAD_BED) and the distributed ice volume in 2020 (MAD_V_2020). Since the simulation runs dynamically from 1980 to 2020 and the distributed volume in 1980 is one of the control variables, we also calculate the MAD of the distributed ice volume in 1980 (MAD_V_1980). Although we do not expect to accurately reconstruct the 1980 glacier state, given the lack of target observations before 2000 and the inherently diffusive nature of glacier dynamics, tracking MAD_V_1980 still offers valuable insight into how strongly diffusion limits the ability to recover past glacier states.

### 4 Results and Discussion

We begin by evaluating the performance of the two first guess methods in Sect. 4.1, with a particular focus on the glacier bed. As noted in Sect. 2.5, a temporal mismatch is introduced in the first guess of the 1980 volume distribution, which limits the interpretability at this stage.

This is followed by an in-depth analysis of the synthetic Aletsch glacier geometry in a retreating dynamic state (Sect. 4.2). This includes an evaluation of AGILE's core functionality (Sect. 4.2.1), the impact of different cost function settings (Sect. 4.2.2), and the influence of various first-guess methods (Sect. 4.2.3).

Finally, we generalize our findings across the different dynamic states and glacier geometries used in the experiments (Sect. 4.3). In Sect. 4.3.1, we assess AGILE's core performance using a fixed $\lambda$ value of 0.01. Sect. 4.3.2 investigates the recovery of



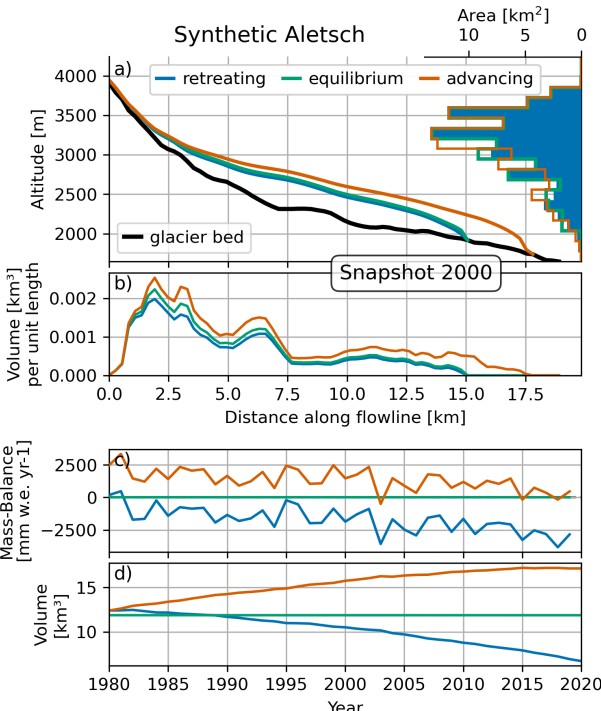

**Figure 2.** Synthetic glacier geometry inspired by Aletsch, shown for retreating, equilibrium, and advancing glacier states. Panel (a) displays the defined glacier bed, ice thickness along the flowline, and area-height distribution for the year 2000. Panel (b) presents the volume distribution along the flowline for the year 2000. Panel (c) illustrates the driving mass balance used during the simulation from 1980 to 2020, and panel (d) shows the corresponding total glacier volume over the same period.

control variables along the flowline. Lastly, in Sect. 4.3.3, we analyze the impact of various cost function settings, including
different $\lambda$ values and varying numbers of target observations, across all glacier geometries for both retreating and advancing cases.

## 4.1 First guess performance

Using our synthetic glacier measurements, we evaluated the two first guess methods described in Sect. 2.5. At this stage, we focus only on the glacier bed, as the use of a 2000-based distributed volume estimate to represent the 1980 distribution
introduces a temporal mismatch (see Sect. 2.5). In later analyses, however, we return to the 1980 volume distribution to assess AGILE's ability to reconstruct past glacier states prior to the first available observation.

     Starting from OGGM's first guess glacier bed, Figure 3 shows the best performance in the equilibrium state, with a maximum absolute difference of 39.5 m and a MAD of 1.5 m. This is expected, as OGGM's inversion assumes the glacier is in equilibrium. In contrast, bed elevations are systematically underestimated in the retreating state and overestimated in the advancing state,
especially near the terminus. These states show maximum absolute differences of 56.1 m (retreating) and 70.5 m (advancing),





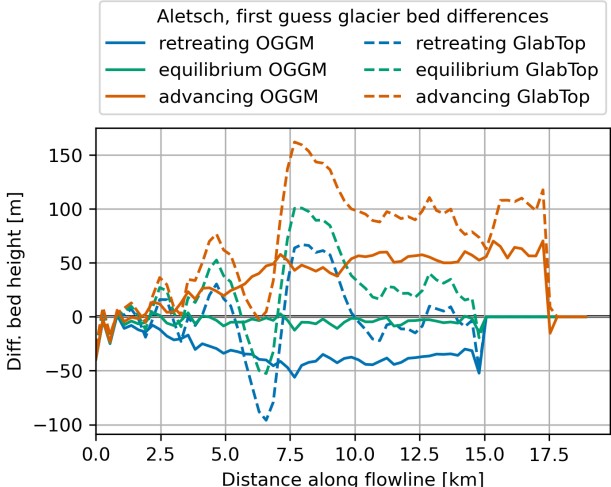

**Figure 3.** Difference between the synthetic glacier bed for Altesch and the first guess glacier bed for the two first guess methods, OGGM and GlabTop (see Sect. 4.1), shown for all three dynamic states: retreating, equilibrium, and advancing.

with MAD values of 8.7 m and 12.7 m, respectively. This pattern holds across all glacier geometries and reflects previous findings (e.g., Figure 5, panel d in Maussion et al., 2019), highlighting the importance of assessing performance under varying dynamic conditions.

In contrast, the first guess glacier bed of the GlabTop method shows similar behavior across all dynamic states, as it relies
solely on surface geometry and does not incorporate any mass balance forcing. The maximum absolute differences for Aletsch in Figure 3 are 96.0 m (retreating), 100.9 m (equilibrium), and 162.2 m (advancing). This behavior is consistent across all tested geometries.

For all other glacier geometries and dynamic states the performance metrics of the two first guess methods are summarized in Table S1, using mean absolute differences for bed height (MAD_BED), distributed volume in 1980 (MAD_V_1980), and
in 2020 (MAD_V_2020). The MAD_V_2020 metric is computed after running the glacier evolution model from 1980 to 2020 with the prescribed mass balance. The OGGM values listed in the table are later used to normalize performance metrics, allowing for easier assessment of whether AGILE improves upon the OGGM first-guess results.

Across all cases, Table S1 shows that OGGM generally performs as well as or better than GlabTop, with the most accurate results seen in equilibrium states. This trend is particularly pronounced for the Baltoro glacier, where OGGM outperforms
GlabTop by nearly an order of magnitude. Consequently, Baltoro represents an especially relevant case for evaluating AGILE's ability to improve upon poor initial guesses that deviate strongly from the synthetic truth.





## 4.2 Aletsch retreating

### 4.2.1 Proof of concept for AGILE functionality

The first experiment showcases the proper functionality of AGILE and illustrates that gradient calculations are working as
expected. For this we selected our synthetic Aletsch-retreating case and set $\lambda$ to 0.01 (see equation 1). Further we use all three target observations we took during the glacier creation, as defined in Sect. 3.1 ($sfc_{2000}$, $V_{2000}$ and $\Delta M_{2000/2020}$) and our first guess is coming from OGGM. Our control variables consist of the distributed volume in 1980 as well as the bed height at each grid point, which gives in total 120 control variables.

Figure 4 panel a shows the evolution of the cost function over 20 iterations, with the largest decrease occurring in the first
two iterations, reducing the cost from 2.3 to 0.2. This demonstrates that the gradient-informed updates to the 120 control variables are working as intended. The correctness of the gradient calculations, along with the simultaneous updating of all control variables, is further highlighted by AGILE requiring only 21 forward model runs for 20 iterations (see Figure 4 panel c), with most iterations needing just one run to find a smaller cost value. This indicates that the minimization algorithm rarely performs 'search' runs, where updates to control variables fail to reduce the cost.

Examining individual cost components (Figure 4, panels a and b), the largest initial contributor is the mismatch to observed surface heights along the flowline $sfc_{2000}$, followed by the volume $V_{2000}$ and the geodetic mass balance $\Delta M_{2000/2020}$ terms. The $sfc_{2000}$ mismatch significantly decreased after two iterations, becoming smaller than $\Delta M_{2000/2020}$, but later increasing slightly again. This highlights that not all observation mismatches decrease monotonically and may temporarily increase as the total cost continues to decrease. The regularization term, starting at 0.005, remained almost constant throughout the iterations.
However, as the total cost decreases, its relative importance grows, eventually becoming the dominant cost component after 13 iterations (Figure 4, panel b).

Analyzing the performance metrics (MAD_BED, MAD_V_1980, and MAD_V_2020) over iterations (Figure 4 panel c) confirms that reductions in the cost function led to improved agreement with the synthetic truth. Significant improvements were observed for both the bed height and the distributed volume in 2020 compared to the first guess. While the distributed
volume for 1980 initially worsened during the early iterations, it began improving relative to the first guess after the sixth iteration.

Figure 5 shows how differences between the modeled and synthetic truth evolved along the flowline, in particular panel a difference of the bed height, panel b difference of the distributed volume 1980 and panel c difference of the distributed volume 2020. Improvements are evident across most of the flowline, except near the highest points (start of the flowline, at 0 distance).
At these points, a persistent noisy pattern is observed. However, this pattern is confined to the initial few grid points, with the majority of the flowline showing substantial improvements.

This experiment demonstrates AGILE's ability to adapt 120 control variables in just a few iterations, achieving significant improvements over the OGGM-provided first guess. Additionally, the computational demand is minimal, with each iteration (forward model run and gradient calculation) taking approximately 1.3 seconds for the Aletsch geometry on a laptop equipped
with an 11th Gen Intel(R) Core(TM) i7-1165G7 CPU (2.8 GHz) and 16 GB RAM (see Table S2 for other computing times of





other dynamic states and geometries). The gradient calculation, specifically the backward propagation through the computing graph (see Appendix A1), requires only around 0.1 seconds, or roughly ten percent of the total iteration. This efficiency makes the method highly promising for regional to global-scale applications.

### 4.2.2 Different cost function settings

Since the cost function is central to AGILE, we tested various configurations to evaluate its performance. Specifically, we explored 30 different values of $\lambda$, including 0 (no regularization) and 29 values ranging logarithmically from $10^{-4}$ to $10^{3}$. Additionally, we varied the number of provided target observations ($sfc_{2000}$, $V_{2000}$ and $\Delta M_{2000/2020}$) across seven configurations: using only one of the three target observations, all combinations of two target observations, and all three target observations together. The performance metrics (MAD_BED and MAD_V_2020) for Aletsch retreating and advancing, using

OGGM for the first guess, after 20 iterations are shown in Figure 6.

  When varying $\lambda$, we observed small improvements when transitioning from no regularization ($\lambda = 0$) to small values of $\lambda$. However, as $\lambda$ increased further, performance worsened. This indicates that overly large regularization weights cause AGILE to prioritize smoothing the bed over matching target observations. It is important to note that in this experiment, we used perfectly accurate target observations. With real, imperfect data, regularization becomes more crucial to prevent overfitting,

which explains why $\lambda = 0$ performed well under these idealized conditions.

  When using only a single target observation type, $sfc_{2000}$ yielded the best results across the widest range of $\lambda$ values. This is expected, as surface height provides distributed information about the glacier's slope along the flowline, while $V_{2000}$ and $\Delta M_{2000/2020}$ are integrated over the entire glacier. Interestingly, even with only providing $V_{2000}$, small $\lambda$ values improved MAD_V_2020.

Combining two target observations consistently outperformed single target observations, especially when $sfc_{2000}$ were included. Notably, combining $V_{2000}$ and $\Delta M_{2000/2020}$ produced significant improvements compared to using either target observation alone. Furthermore, using two target observations broadened the range of $\lambda$ values that yielded improvements over the first guess.

  Using all three target observations simultaneously did not provide substantial benefits compared to configurations that in-

cluded two target observations, particularly when $sfc_{2000}$ was one of them. This suggests that two target observations may already provide sufficient information for the control variables for this particular experiment.

### 4.2.3 Influence of first guess

This section compares AGILE's performance when starting from the OGGM first guess versus the GlabTop first guess. As shown in Figure 4 panels a and d, the initial cost value at iteration 0 is significantly higher for GlabTop (10.1) than for OGGM

(2.3), due to larger mismatches with the target observations ($sfc_{2000}$, $V_{2000}$, and $\Delta M_{2000/2020}$). Despite this, AGILE effectively minimizes the cost function, reaching a comparable value to OGGM's initial cost after three iterations and continuing to decrease thereafter.





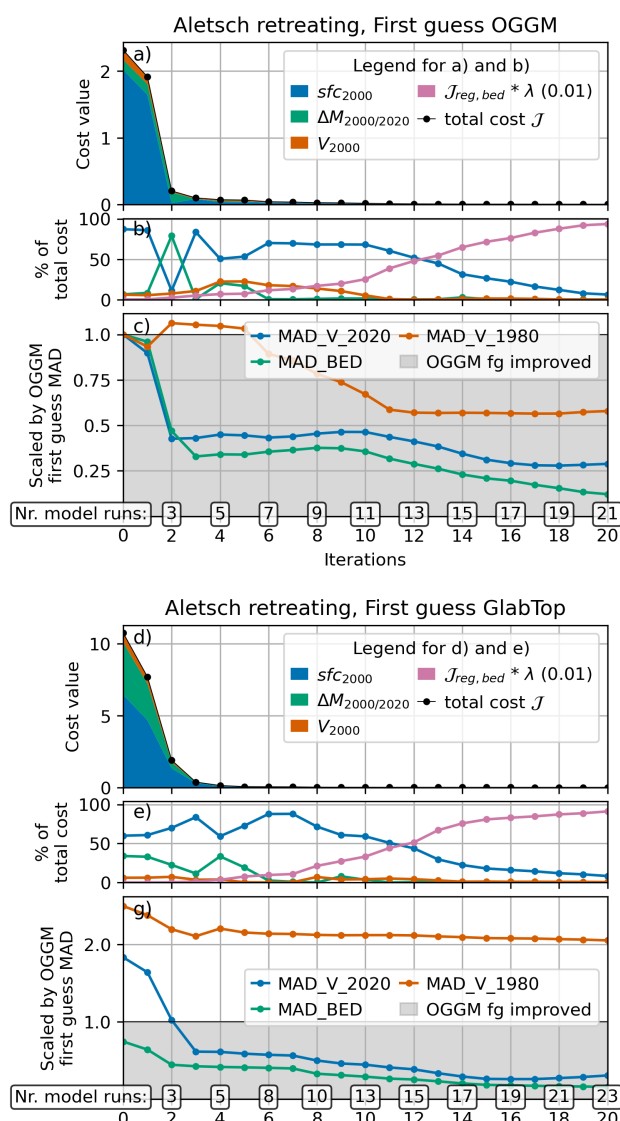

**Figure 4.** Evolution of the cost function and performance metrics for Aletsch retreating, starting from the OGGM first guess (panels a to c) and GlabTop first guess (panels d to f), with a $\lambda$ value of 0.01. Panels a and d shows the evolution of the cost function over minimization iterations, with individual cost terms represented in different colors. The relative contribution of the cost terms to the total cost is shown in panels b and e. Panels c and f illustrates the evolution of performance metrics over the same iterations. The numbers in boxes along the x-axis indicate the total number of forward model runs required for each corresponding iteration.

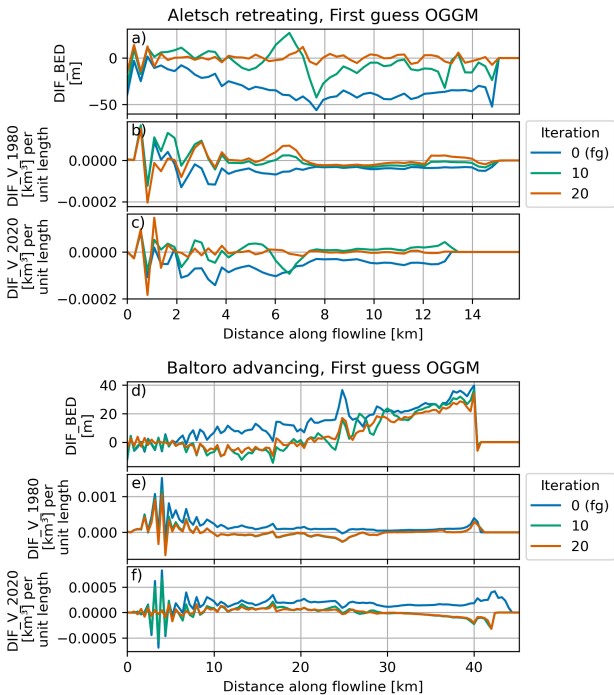

**Figure 5.** Differences between the synthetic truth and AGILE guess after 10 and 20 iterations along the flowline for Aletsch retreating (panels a to c) and Baltoro advancing (panels d to f), starting from the OGGM first guess (0 Iteration) with a $\lambda$ value of 0.01. Panels a and d shows the differences in bed height, panels b and e shows the differences in volume for 1980, and panels c and f shows the differences in volume for 2020. All differences are displayed for every grid point along the flowline.

Examining the performance metrics (Figure 4 panels g), we find that at Iteration 0, GlabTop has a slightly better MAD_BED compared to OGGM, while MAD_V_1980 and MAD_V_2020 are approximately twice as large. With further iterations, all three metrics improve. After three iterations, MAD_V_2020 improves enough to outperform the OGGM first guess (value < 1). However, MAD_V_1980 shows only minor improvement after 20 iterations and remains roughly twice as large as the value achieved with OGGM.

These results highlight that starting from a worse first guess, such as GlabTop, can reduce the accuracy of reconstructing the glacier state in 1980. This limitation is partly due to the diffusive nature of glacier dynamics, which leads to a gradual loss of information over time during dynamic simulations. Nonetheless, AGILE demonstrates its ability to improve even a poor first guess, successfully refining the glacier bed and simultaneously initialize the model with the 2020 dynamic glacier state.

## 4.3 Influence of glacier geometry and dynamic state

To evaluate the impact of glacier geometry and the dynamic state, we generated 12 synthetic glaciers representing four geometries (Aletsch, Artesonraju, Baltoro, and Peyto) in three dynamic states: retreating, equilibrium, and advancing, as outlined





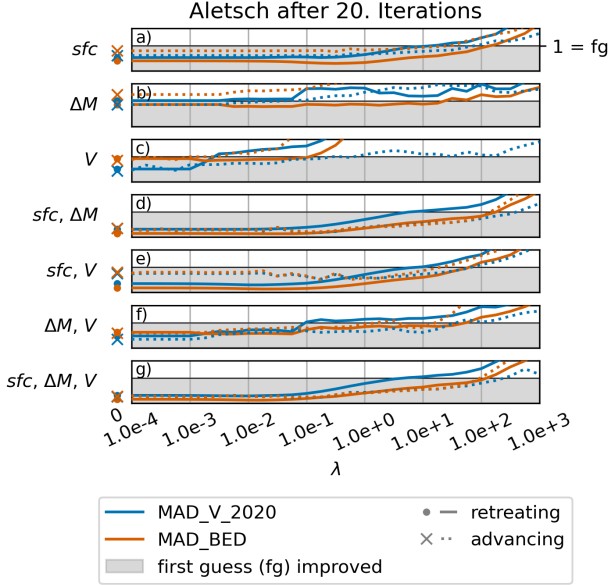

**Figure 6.** MAD_BED and MAD_V_2020, normalized by the OGGM first guess values, for Aletsch retreating (dot and solid line) and advancing (cross and dotted line) after 20 iterations, starting from the OGGM first guess. The y-axis shows the number of provided target observations, and the x-axis shows different values of $\lambda$ ($\lambda = 0$ is shown with a dot or a cross to the left of the axis). The gray shaded are indicates the region where the OGGM first guess could be improved after 20. Iterations.

in Sect. 3.1. First, we examine the core functionality of minimizing the cost across these 12 settings in Sect. 4.3.1. Next, we analyze how the differences between the synthetic truth and the estimates of AGILE along the flowline evolve with iterations in Sect. 4.3.2. Finally, we investigate the effects of different cost function settings in Sect. 4.3.3.

### 4.3.1 Functionality of minimizing the cost

In the equilibrium experiments, the OGGM first guess performs very well, which is expected given its underlying equilibrium

assumption. The only exception is Baltoro, where the first guess for the equilibrium case closely resembles that of the retreating case. As a result, the minimization process also mirrors the retreating scenario.

As an example of typical behavior in the equilibrium case, Figure 7 panels a–c show the results for the Aletsch geometry starting from the OGGM first guess. The strong performance of the OGGM first guess is evident in the low initial total cost of just 0.18 (panel a), indicating that mismatches for all observations fall within their defined uncertainty ranges, suggesting that

further optimization is not strictly necessary. Nonetheless, AGILE is able to refine the solution and improve upon this already strong first guess, as shown by the decreasing values of the performance metrics in panel c.

To assess AGILE's performance when starting from a first guess that significantly deviates from the synthetic truth, Figure 7, panels d–e, shows the results for the Baltoro geometry initialized with the GlabTop first guess. The poor initial performance





is evident from the high total cost of 82.5 (panel d) and large values of the performance metrics, around 10, which means they

are approximately ten times higher than those for the OGGM first guess.

Despite this challenging starting point, AGILE is able to substantially improve the solution. After 20 iterations, it reaches performance levels close to those of the OGGM first guess, with values around 1.4 (where 1 indicates equal performance to the OGGM first guess). When AGILE is allowed to continue beyond 20 iterations, it further improves the results, achieving values around 0.5 for both MAD_BED and MAD_V_2020 after 40 iterations (not shown in panel e). However, MAD_V_1980 does

not improve further beyond 20 iterations, highlighting the limited recoverability of the 1980 glacier state.

This pattern, where MAD_BED and MAD_V_2020 improve more significantly than MAD_V_1980, is observed in many experiments. For example, Figure 8, panel c (Artesonraju, advancing) and panel g (Peyto, retreating), both starting from the OGGM first guess, show similar behavior. This highlights the broader challenge of reconstructing past glacier states in the absence of direct observations, regardless of glacier geometry or dynamic state. As discussed earlier, this limitation stems from

the diffusive nature of glacier dynamics and the resulting loss of information over time.

For all glacier geometries and dynamic states, AGILE required 21 to 27 model runs over 20 iterations, demonstrating its ability to efficiently inform the minimization algorithm with accurate gradients. An exception was the retreating Peyto experiment (Figure 8, panel g), where the number of model runs increase a lot from Iteration 14 ongoing. This hints that the minimisation algorithm has difficulties in further minimising the cost as probably a local minimum is reached. For real-world

applications, AGILE includes an option to limit the number of forward model runs (default: 100) to reduce computational effort. Forward model run times varied between 0.7 and 2.5 seconds (see Table S2), depending on the maximum glacier velocity and corresponding time-step constraints imposed by the stability criterion (Equation A19).

### 4.3.2 Distributed differences along the flowline

Examining differences along the flowline for bed height (DIF_BED), and distributed volumes in 1980 (DIF_V_1980) and

2020 (DIF_V_2020) reveals improvements at most grid points, regardless of dynamic state or first-guess method. However, one notable exception occurs in the Baltoro advancing case (Figure 5, panels d–f), where AGILE was unable to correct an overly high glacier bed at the terminus. Despite this limitation, clear improvements are observed between 5–25 km along the flowline. As a large portion of the glacier's total volume is located within this section (Figure S1, panel f), the distributed volume is still significantly improved.

Noisy patterns in flowline differences appear in several cases, such as Aletsch retreating and Baltoro advancing (Figure 5). With increasing iterations, these patterns seen in DIF_BED, are either dampened or remain stable in DIF_V_2020, and occasionally are amplified in DIF_V_1980. The variability depends on the corresponding glacier volume along the flowline (Figure 2, panel b; Figure S1, panels b, f, and j). These patterns likely reflect the diffusive nature of glacier dynamics and the inherent limitations in accurately inverting for distributed glacier volume when relying on glacier-integrated observations.





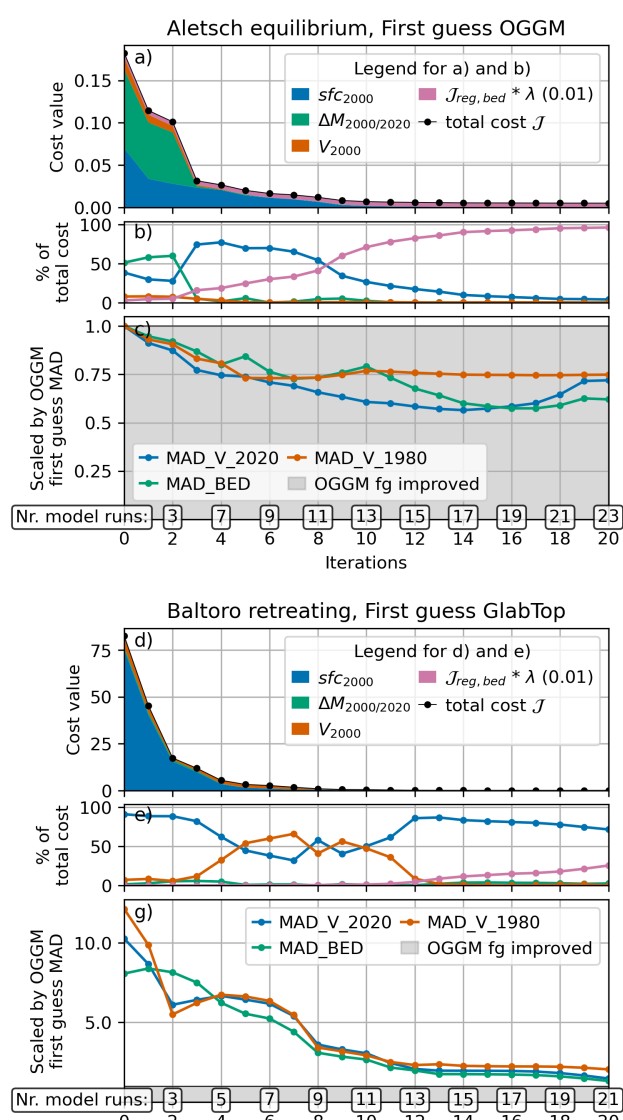

**Figure 7.** Same as Figure 4, for Aletsch (a, b and c) equilibrium using OGGM first guess, and Baltoro (d, e and f) equilibrium using GlabTop first guess.



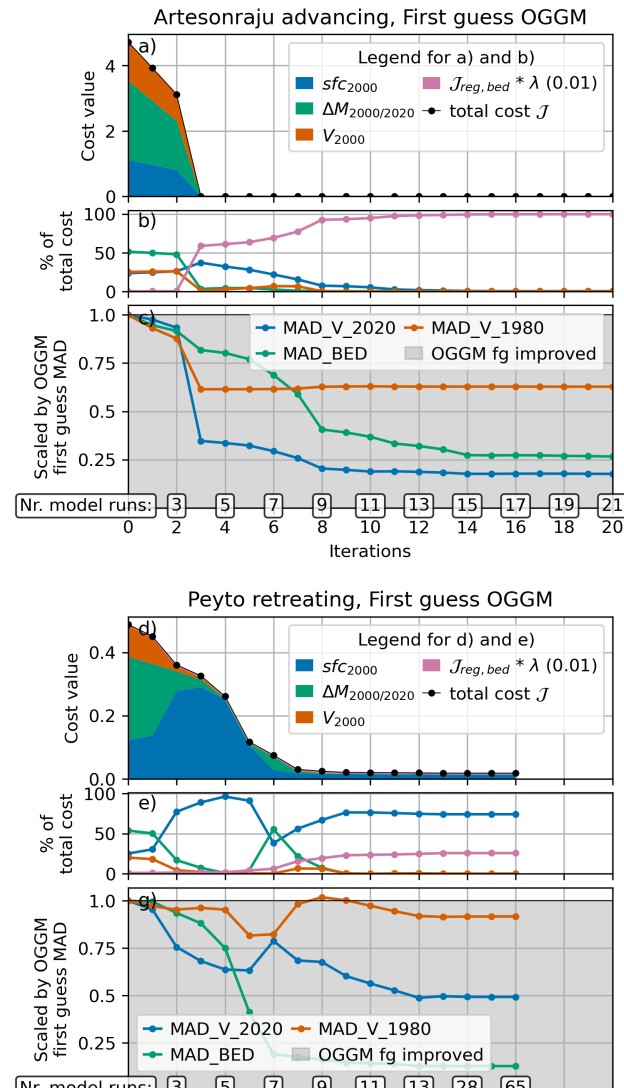

**Figure 8.** Same as Figure 4, for Artesonraju advancing (a, b and c) and Peyto retreating (d, e and f) starting from the OGGM first guess.





### 4.3.3 Performance Across Cost Function Settings

We analyze here the impact of different cost function settings ($\lambda$ values and number of provided target observations) for retreating and advancing cases, focusing on individual glacier geometries. All experiments analysed start from the OGGM first guess and we evaluate the performance depending if the first guess can be improved by looking at MAD_BED and MAD_V_2020. Additionally, we consider the range of $\lambda$ values that result in improved performance. In general, larger $\lambda$ values increase the regularization term, favouring a smoother glacier bed. When $\lambda$ becomes too large, the cost function prioritizes bed smoothing over matching the target observations, leading to worse performance.

For Aletsch, in both the retreating and advancing cases, surface height observations ($sfc_{2000}$) had the strongest influence in improving MAD_BED and MAD_V_2020 compared to the OGGM first guess (Figure 6). Among the two-observation combinations, the pairing of $sfc_{2000}$ and $\Delta M_{2000/2020}$ yielded the best performance. In the advancing case, this was followed by $\Delta M_{2000/2020}$ and $V_{2000}$, while in the retreating case, $sfc_{2000}$ and $V_{2000}$ was the next best pair. The largest improvements overall were achieved when all three observations were used together.

For Artesonraju (Figure 9, panels a to g), $sfc_{2000}$ was by far the most important target observation. Even when used alone, it achieved results comparable to using two or all three observations. Combining $V_{2000}$ and $\Delta M_{2000/2020}$ did not outperform using these observations individually.

For Baltoro (Figure 9, panels h to n), in the retreating case (solid lines) single observations alone were insufficient to improve upon the first guess, with $V_{2000}$ showing the best performance. Combining $V_{2000}$ with either $sfc_{2000}$ or $\Delta M_{2000/2020}$ resulted in substantial improvements, with the best performance achieved using all three target observations. For the advancing case (dotted lines), $sfc_{2000}$ alone was sufficient to improve MAD_BED and MAD_V_2020, with further gains observed when combining target observations.

In the retreating case for Peyto (solid lines in Figure 9, panels o to u), simultaneous improvement of MAD_BED and MAD_V_2020 was only achieved by using all three target observations. For other configurations, improving one metric often led to no improvement or degradation in the other. In the advancing case (dotted lines), $sfc_{2000}$ alone improved both metrics, with further improvements observed when combining target observations, though using all three added little additional value.

The importance of specific target observations varies with glacier geometry and dynamic state. While it is challenging to disentangle the roles of geometry and state, increasing the number of target observations consistently improves performance. All cases benefit from using all three target observations, though in some instances, performance with one or two target observations is nearly equivalent.

Further, the results demonstrate that the range of effective $\lambda$ values is relatively broad, with many retreating and advancing cases performing well for $\lambda$ values between $10^{-4}$ and $10^{-1}$. The use of $\lambda = 0$ also performs effectively in our experiments with perfectly accurate measurements, as overfitting is not a concern. However, in real-world applications where observations contain uncertainties, the regularization term becomes critical for preventing such an overfitting.





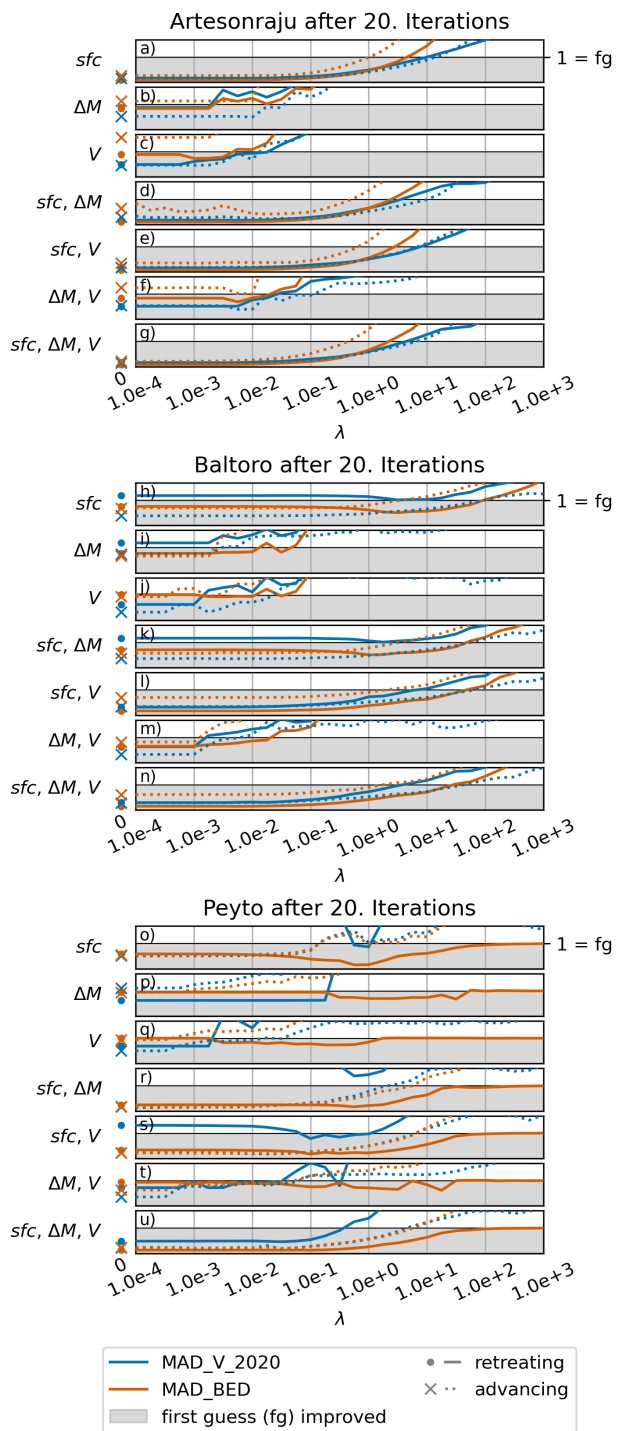

**Figure 9.** Same as Figure 6 for Artesonraju (panels a to g), Baltoro (panels h to n), and Peyto (panels o to u).



## 5 Conclusion and future work

Our experiments show that AGILE can adjust many control variables in just a few iterations, proving that the gradient calculations using AD work well. Adding more target observations makes the inversion more reliable, but in some cases, good results can be achieved with fewer target observations. Reconstructing the glacier volume in 1980 was harder than in 2020 because no direct target observations were used before 2000, and the diffusive nature of glacier dynamics, which leads to a gradual loss of information over time during dynamic simulations.

These results are promising, but AGILE has not yet been tested on real-world data. Its flexibility offers many possibilities, but these need careful exploration. For example, with new glacier bed estimates becoming available (e.g., Cook et al., 2023; van Pelt and Frank, 2025), AGILE could focus on other aspects, like combining the calibration of glacier dynamics and mass balance models. Setting up test cases with plenty of observations for validation will be key to understanding what AGILE can do in practical applications.

In our idealized experiments, we assumed the target observations were perfect, so regularization was less important. In real-world cases, where observations are uncertain, regularization will be critical to avoid overfitting. Finding the right balance for the regularization weight ($\lambda$) will be necessary. Techniques like L-curves (see, e.g., Hansen, 1992; Gillet-Chaulet et al., 2012; Recinos et al., 2023; Wolovick et al., 2023), could help in deciding this balance.

The next step for applying AGILE to real-world problems would be to include a differentiable mass balance model. A simple temperature index model (e.g., Marzeion et al., 2012) could be a good starting point. For more complex mass balance models (e.g. PyGEM Rounce et al., 2020) it will be important to check if their equations work with AD. Similarly, adding dynamic processes like calving or debris cover will require compatibility with AGILE's AD framework. Therefore, AD may limit how complex the models can get, which is further restricted by the availability of target observations. On the other hand, AGILE's ability to work with diverse datasets could support more complex models as more target observations become useable in a consistent way.

AGILE provides a promising way to integrate new datasets as they become available or to use all available glacier-specific observations in a consistent way. This could help address the issue of equifinality in global glacier modeling (e.g. Rounce et al., 2020) by combining all available information. Additionally, AGILE could generate consistent glacier histories, filling in data gaps from the past (e.g., creating a reanalysis dataset for glaciers) or providing a solid starting point for future projections.

*Code availability.* AGILE is written in Python and openly available on GitHub (https://github.com/OGGM/AGILE, last access: 14 July 2025) under a BSD 3-Clause License. Version 0.1 is also archived on Zenodo with a permanent DOI (Schmitt et al., 2025). To ensure full reproducibility, all scripts used to run the experiments and generate the figures in this study are included in the same repository under https://github.com/OGGM/AGILE/tree/v0.1.1/agile1d/sandbox/paper_v01_code (last access: 14 July 2025). In addition, we provide Docker images of the computing environment used, publicly accessible via GitHub Packages (https://github.com/OGGM/AGILE/pkgs/container/agile, last access: 14 July 2025). All results presented in this work were produced using the Docker image tagged agile:20230525.





**Appendix A: Model implementations using PyTorch**

**A1    Automatic Differentiation in PyTorch**

PyTorch implements automatic differentiation (AD) using a system called Autograd, which is designed for deep learning optimization tasks. This system enables the construction of a dynamic computation graph and efficient gradient computation. We use this system in our forward model by replacing NumPy arrays with PyTorch tensors.

During a forward pass, PyTorch constructs a dynamic computational graph that keeps track of all operations on tensors

requiring gradients, in our case the control variables. For those, all operations involving this tensor are tracked for later differentiation.

After the forward pass, we call PyTorch's .backward() function on the cost function tensor. This triggers PyTorch to compute the gradient of that tensor with respect to all tensors requiring gradients, in our case the partial derivatives of our control variables with respect to the cost function. This computation is done by propagating the computational graph backward and

applying the chain rule

$$\frac{dy}{dz} = \frac{dy}{dx}\frac{dx}{dz} \tag{A1}$$

for each operation applied during the forward pass. To achieve this, PyTorch stores the derivatives of all standard operations.

PyTorch also allows users to add custom operations by defining both a forward computation and a corresponding backward computation (the derivative of the forward computation). This can be useful for optimizing parts of the code where the

derivatives are known. We applied this approach in our Semi-Implicit solver (see Sect. A2).

**A2    Semi-Implicit Solver including AD**

With the idea of a global application in mind, AGILE uses the same 1.5D flowline representation as OGGM. In particular, AGILE uses one flowline with changing widths. The bed shape is trapezoidal, with a constant wall angle of 45°. The flowlines are generated from the geographical input date using the 'elevation band flowlines' method (e.g. Huss and Farinotti, 2012;

Huss and Hock, 2015; Werder et al., 2019).

The glacier evolution model from OGGM (originally introduced by Oerlemans, 1997)

$$\frac{\partial C}{\partial t} = w\dot{m} - \nabla \cdot q \tag{A2}$$

is adapted in AGILE and re-implemented with PyTorch. This advection equation for the cross-section area $C$ (m$^2$) provides flexibility for varying surface widths $w$ (m) and allows the use of mixed bed shapes with the same equation. Further $\dot{m}$ is the

mass balance (kg m$^{-2}$ s$^{-1}$), $q = Cu$ is the ice flux (m$^3$ s$^{-1}$) and $u$ is the depth-integrated shallow-ice velocity (m s$^{-1}$). This velocity is defined as





$$u = \left( \frac{2A}{n+2} h + f_s \frac{1}{h} \right) \left( -\rho g h \frac{\partial s}{\partial x} \right)^n \tag{A3}$$

where $A$ is the ice creep parameter (s$^{-1}$ Pa$^{-3}$), $n$ is the exponent of Glen's flow law ($n = 3$), $h$ is the ice thickness (m), $f_s$ is a sliding parameter (s$^{-1}$ Pa$^{-3}$), $\rho$ is the ice density (900 kg m$^{-3}$), $g$ is the gravitational acceleration (9.81 m s$^{-2}$) and $\frac{\partial s}{\partial x}$ is

the surface slope, where $s$ is the surface height (m). By default, the sliding parameter $f_s$ is set to 0 s$^{-1}$ Pa$^{-3}$ because there are currently no methods/observations available on a global scale to distinguish the contributions of ice deformation (defined by $A$) and sliding to the total velocity.

Besides the explicit forward finite difference approximation scheme of OGGM also a semi-implicit scheme is included in AGILE. The reason for this is that numeric instabilities can occur (a known problem, see e.g. https://oggm.org/2020/01/18/

stability-analysis/ and https://oggm.org/2020/07/08/numerics/) which cause problems when using AD. In particular, during the backward pass for the gradient calculation, these instabilities are amplified and dominate the resulting gradients. The semi-implicit scheme was derived by starting from equation A2 and rearranging it from an advection equation into a diffusion equation

$$\frac{\partial C}{\partial t} = w\dot{m} + \nabla \cdot (D \frac{\partial s}{\partial x}) \tag{A4}$$

with Diffusivity D

$$D = \left( \frac{2A}{n+2} h + f_s \frac{1}{h} \right) (\rho g h)^n |\frac{\partial s}{\partial x}|^{n-1} C. \tag{A5}$$

In the following the derivation of the semi-implicit scheme using a rectangular cross-section $C = wh$ is shown. This simplifies the problem because for a rectangular bed shape the surface width is not changing over time ($\frac{\partial C}{\partial t} = \frac{\partial wh}{\partial t} = w\frac{\partial h}{\partial t}$). Afterwards, this solution is generalised for a trapezoidal cross-section. First, we modify equation A4 by using the rectangular

cross-section area

$$\frac{\partial h}{\partial t} = \dot{m} + \frac{1}{w} \nabla \cdot (D \frac{\partial s}{\partial x}) \tag{A6}$$

with Diffusivity $D$

$$D = \left( \frac{2A}{n+2} h^2 + f_s \right) (\rho g h)^n |\frac{\partial s}{\partial x}|^{n-1} w. \tag{A7}$$

For the discretization, the ice flux $q$ is defined on a staggered grid denoted with indices $i \pm 1/2$. Further using $(\nabla q)_i = \frac{q_{i+1/2} - q_{i-1/2}}{\Delta x}$, $\left( \frac{\partial s}{\partial x} \right)_{i+1/2} = \frac{s_{i+1} - s_i}{\Delta x}$ and $q^t = D^t \left( \frac{\partial s}{\partial x} \right)^{t+1}$ we get






$$\frac{h_i^{t+1} - h_i^t}{\Delta t} = \dot{m} + \frac{1}{w_i} \frac{D_{i+1/2}^t \left(\frac{s_{i+1}^{t+1} - s_i^{t+1}}{\Delta x}\right) - D_{i-1/2}^t \left(\frac{s_i^{t+1} - s_{i-1}^{t+1}}{\Delta x}\right)}{\Delta x}. \tag{A8}$$

For spatial interpolation of variables from the unstaggered to the staggered grid an arithmetic mean value is used. This is needed in the calculation of $D$ for $h$ and $w$ (e.g. $h_{i+1/2} = \frac{h_i + h_{i+1}}{2}$).

Next, we rearrange the equation to put all terms involving the future time-step $t+1$ on the left side

$$h_i^{t+1} - \frac{\Delta t}{\Delta x^2 w_i}\left(D_{i-1/2}^t s_{i-1}^{t+1} - (D_{i+1/2}^t + D_{i-1/2}^t)s_i^{t+1} + D_{i+1/2}^t s_{i+1}^{t+1}\right) = h_i^t + \Delta t \dot{m}. \tag{A9}$$

and use the definition of the surface height $s = b + h$ (where the glacier bed height $b$ is constant over time) together with a vector notation

$$\left[-\frac{\Delta t}{\Delta x^2 w_i}D_{i-1/2}^t \quad 1 + \frac{\Delta t}{\Delta x^2 w_i}(D_{i+1/2}^t + D_{i-1/2}^t) \quad -\frac{\Delta t}{\Delta x^2 w_i}D_{i+1/2}^t\right] \cdot \begin{bmatrix} h_{i-1}^{t+1} \\ h_i^{t+1} \\ h_{i+1}^{t+1} \end{bmatrix} =$$
$$h_i^t + \Delta t \dot{m} + \frac{\Delta t}{\Delta x^2 w_i}\left[D_{i-1/2}^t \quad -(D_{i+1/2}^t + D_{i-1/2}^t) \quad D_{i+1/2}^t\right] \cdot \begin{bmatrix} b_{i-1} \\ b_i \\ b_{i+1} \end{bmatrix}. \tag{A10}$$

Finally, this equation can be used to set up a final linear equation for all grid points. We include the boundary conditions
$D_{-1/2} = D_{nx+1/2} = 0$ for all time steps, where $nx$ denotes the last grid point of the unstaggered grid and the unstaggered grid starts with index 0. With this, we define two $nx \times nx$ matrices





$$
\boldsymbol{M}_h = \boldsymbol{\delta}_{i+1,j} \cdot
\begin{bmatrix}
-\frac{\Delta t}{\Delta x^2 w_0} D_{-1/2} \\
-\frac{\Delta t}{\Delta x^2 w_1} D_{1/2} \\
\vdots \\
-\frac{\Delta t}{\Delta x^2 w_{nx}} D_{nx-1/2}
\end{bmatrix} +
$$

$$
+ \boldsymbol{\delta}_{i,j} \cdot
\begin{bmatrix}
1 + \frac{\Delta t}{\Delta x^2 w_0}(D_{-1/2} + D_{1/2}) \\
1 + \frac{\Delta t}{\Delta x^2 w_1}(D_{1/2} + D_{3/2}) \\
\vdots \\
1 + \frac{\Delta t}{\Delta x^2 w_{nx}}(D_{nx-1/2} + D_{nx+1/2})
\end{bmatrix} +
$$

$$
+ \boldsymbol{\delta}_{i,j+1} \cdot
\begin{bmatrix}
-\frac{\Delta t}{\Delta x^2 w_0} D_{1/2} \\
-\frac{\Delta t}{\Delta x^2 w_1} D_{3/2} \\
\vdots \\
-\frac{\Delta t}{\Delta x^2 w_{nx}} D_{nx+1/2}
\end{bmatrix} \tag{A11}
$$

and

$$
\boldsymbol{M}_b = \boldsymbol{\delta}_{i+1,j} \cdot
\begin{bmatrix}
\frac{\Delta t}{\Delta x^2 w_0} D_{-1/2} \\
\frac{\Delta t}{\Delta x^2 w_1} D_{1/2} \\
\vdots \\
\frac{\Delta t}{\Delta x^2 w_{nx}} D_{nx-1/2}
\end{bmatrix} +
$$

$$
+ \boldsymbol{\delta}_{i,j} \cdot
\begin{bmatrix}
-\frac{\Delta t}{\Delta x^2 w_0}(D_{-1/2} + D_{1/2}) \\
-\frac{\Delta t}{\Delta x^2 w_1}(D_{1/2} + D_{3/2}) \\
\vdots \\
-\frac{\Delta t}{\Delta x^2 w_{nx}}(D_{nx-1/2} + D_{nx+1/2})
\end{bmatrix} +
$$

$$
+ \boldsymbol{\delta}_{i,j+1} \cdot
\begin{bmatrix}
\frac{\Delta t}{\Delta x^2 w_0} D_{1/2} \\
\frac{\Delta t}{\Delta x^2 w_1} D_{3/2} \\
\vdots \\
\frac{\Delta t}{\Delta x^2 w_{nx}} D_{nx+1/2}
\end{bmatrix} \tag{A12}
$$

555   where $\boldsymbol{\delta}$ is the Kronecker delta defined as

$$
\boldsymbol{\delta}_{i,j} =
\begin{cases}
0 \text{ if } i \neq j \\
1 \text{ if } i = j
\end{cases} \tag{A13}
$$





and

$$\boldsymbol{\delta}_{i+1,j} = \begin{bmatrix} 0 & 0 & \cdots & 0 & 0_{nx} \\ 1 & 0 & \cdots & 0 & 0 \\ 0 & 1 & \cdots & 0 & 0 \\ \vdots & \vdots & \ddots & \vdots & \vdots \\ 0 & 0 & \cdots & 1 & 0_{nx} \end{bmatrix}, \boldsymbol{\delta}_{i,j+1} = \begin{bmatrix} 0 & 1 & 0 & \cdots & 0_{nx} \\ 0 & 0 & 1 & \cdots & 0 \\ \vdots & \vdots & \vdots & \ddots & \vdots \\ 0 & 0 & 0 & \cdots & 1 \\ 0 & 0 & 0 & \cdots & 0_{nx} \end{bmatrix}.$$

The final linear equation we solve is defined as

$$\boldsymbol{M}_h \cdot \begin{bmatrix} h_0^{t+1} \\ h_1^{t+1} \\ \vdots \\ h_{nx}^{t+1} \end{bmatrix} = \begin{bmatrix} h_0^t \\ h_1^t \\ \vdots \\ h_{nx}^t \end{bmatrix} + \Delta t \dot{m} + \boldsymbol{M}_b \cdot \begin{bmatrix} b_0 \\ b_1 \\ \vdots \\ b_{nx} \end{bmatrix} \tag{A14}$$

which can be solved by using the function scipy.linalg.solve_banded from the Python package SciPy (Virtanen et al., 2020).

This solution for a rectangular cross-section can be generalized to a trapezoidal cross-section by using the definition of the area

$$C = \frac{1}{2}(w + w_0)h, \tag{A15}$$

where $w_0$ is the bottom width (m) and the surface width $w$ (m) is defined as

$$w = w_0 + \eta h, \tag{A16}$$

where $\eta$ defines the angle of the side wall (e.g. $\eta=2$ is a 45° wall angle). Now we can use this definition of the cross-section area in the definition of the diffusivity (equation A5)

$$D = \left( \frac{2A}{n+2}h^2 + f_s \right) (\rho g h)^n |\frac{\partial s}{\partial x}|^{n-1} \frac{w_0 + w}{2}. \tag{A17}$$

Further we can rewrite $\frac{\partial C}{\partial t}$ by inserting the trapezoidal cross-section (equation A15) together with the surface width definition (equation A16) to get

$$\frac{1}{2}\frac{\partial}{\partial t}(h(2w_0 + \eta h)) = \frac{1}{2}\frac{\partial h}{\partial t}(2w_0 + \eta h) + \frac{1}{2}h\eta\frac{\partial h}{\partial t} = \frac{1}{2}(2w_0 + 2\eta h)\frac{\partial h}{\partial t} = w\frac{\partial h}{\partial t}. \tag{A18}$$



With this we are able to rewrite equation A4 again into equation A6 and use our solution derived before. The only difference is that we need to use the diffusivity of the trapezoidal cross-section defined in equation A17.

For the time-stepping the stability criterion

$$\Delta t <= cfl\_nr \frac{\Delta x^2}{\max(D_{i+1/2}/w_{i+1/2})} \tag{A19}$$

is used. This criterion derives from a linearised form of equation A6 (assuming D is constant), which will become a heat equation with diffusivity $D/w$. Therefore the stability criterion of the heat equation is adapted here. The $cfl\_nr$ was set to 0.5 following Hindmarsh (2001) equation 91.

To use this semi-implicit solver now in AGILE we further need a differentiable version of the linear equation solver scipy.linalg.solve_banded. Otherwise, the linear solve would need too many operations and the computational graph (see Sect. A1 more infos about the computational graph) will become very large and hence the memory consumption when using AD. For this, a new function is defined which uses the original SciPy solver during the forward pass, which solves equation A14 for $\boldsymbol{h}^{t+1}$ by

$$\boldsymbol{h}^{t+1} = \boldsymbol{M}_h^{-1} \cdot \boldsymbol{rhs} \tag{A20}$$

where

$$\boldsymbol{h}^{t+1} = \begin{bmatrix} h_0^{t+1} \\ h_1^{t+1} \\ \vdots \\ h_{nx}^{t+1} \end{bmatrix} \tag{A21}$$

and

$$\boldsymbol{rhs} = \begin{bmatrix} h_0^t \\ h_1^t \\ \vdots \\ h_{nx}^t \end{bmatrix} + \Delta t \dot{m} + \boldsymbol{M}_b \cdot \begin{bmatrix} b_0 \\ b_1 \\ \vdots \\ b_{nx} \end{bmatrix}. \tag{A22}$$

For the backward pass of this newly defined function, the equations 7 and 8 from Goldberg and Heimbach (2013) are used to calculate the adjoints $\delta^* \boldsymbol{M}_h$ and $\delta^* \boldsymbol{rhs}$ with

$$\delta^* \boldsymbol{rhs} = \boldsymbol{M}_h^{-T} * \delta^* \boldsymbol{h}^{t+1} \tag{A23}$$

and



$$\delta^* \boldsymbol{M}_h = -\delta^* \boldsymbol{rhs} \cdot (\boldsymbol{h}^{t+1})^T \odot (\boldsymbol{\delta}_{i,j+1} + \boldsymbol{\delta}_{i,j} + \boldsymbol{\delta}_{i+1,j}), \tag{A24}$$

where $\odot$ is an element-wise matrix multiplication. This matrix multiplication ensures that we only get gradients of the non-zero elements of $\boldsymbol{M}_h$. The $\delta^*$ notation is used for the adjoints and how they are connected to the gradients is explained in more detail in Goldberg and Heimbach (2013). The implementation of the gradient calculation of this new function was tested against a finite-difference approximation using torch.autograd.gradcheck from Pytorch.

## A3 Mass-Balance Model wrapper

The idea of the wrapper is to circumvent the need to re-implementing part of the modelling chain with PyTorch, but still incorporate its influence on the gradient calculation. The downside is that with this you can not obtain gradients of parameters used in the wrapped part.

In our case, we decided to put the MB model forcing into a wrapper. The MB model gives us for each year a value for the climatic MB depending on the height $mb_{orig}(height)$. The MB height profile is defined each year by the temperature and
precipitation input.

The idea is to utilize the differentiable 1d interpolation tool from https://github.com/aliutkus/torchinterp1d. With this, we define constant height bins $h_{bins}$ which cover the vertical glacier extension. The differentiable MB wrapper finally looks like

$$mb_{torch,i}(h) = \text{torchinterp1d}(h_{bins}, mb_{orig,i}(h_{bins}))(h) \tag{A25}$$

for a year $i$.

This method works best if the desired modelling part is smooth and does not change too much depending on the input. The spacing of bins ($h_{bins}$ in our case) should be determined by the variability of the output of the wrapped function.

*Author contributions.* P.G. initiated the development of AGILE during his master's thesis, after an idea of F.M. P.S. continued this work in his own master's thesis and subsequently implemented the methods presented in this study. D.G. designed the semi-implicit numeric scheme and P.S. implemented it. P.S. conceptualized the study, designed the idealized experiment setup, and carried out the analysis with continuous
input from all co-authors. P.S. wrote the manuscript with contributions from all authors. Large language models were used to assist with grammar, wording, and style during the preparation of the manuscript.

*Competing interests.* The authors declare that they have no conflict of competing interest. Some authors are members of the editorial board of journal Geoscientific Model Development.



*Acknowledgements.* P.S. and F.M. received funding from the European Union's Horizon 2020 research and innovation programme (grant
agreement no. 101003687) (PROVIDE), from the Austrian Climate Research Programme (ACRP) – 14th call, under grant agreement no.
KR21KB0K00001 (HyMELT-CC), and from ESA's "Digital Twin Component for Glaciers" project (4000146160/24/I-KE). D.N.G. acknowl-
edges funding from NERC grant NE/X005194/1.



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
