# Peer review of "The Open Global Glacier Data Assimilation Framework (AGILE) v0.1"

_EGUsphere, 2025_

## Referee Comment (RC1)

**Review of Schmitt et al. (2025) 'The Open Global Glacier Data Assimilation Framework (AGILE) v0.1'**

**Summary**
This paper presents a new workflow built on OGGM that allows improved inversion of glaciers and the construction of dynamically consistent states suitable for use as initialisation for prognostic simulations. The method is applied to idealised versions of four glaciers as a proof-of-concept and shown to work quickly and effectively in refining the coarse initial guess at the glacier profile derived using two different current methods and then running forward to hit the target 2020 state across all four glaciers and regardless of whether the glaciers are assumed to be advancing, retreating or in dynamic equilibrium.

This is a very thorough, well-written paper that convincingly presents the new workflow. The only thing missing is an application to real cases, but that is very deliberately set outside the scope of this paper by the authors; I only hope that they do shortly follow up with such a study! Certainly, as presented, the method is undeniably effective in the synthetic cases included in this paper. Otherwise, I have only a few comments of a very minor nature, so I recommend the paper be accepted subject to minor revisions. Congratulations to the authors – this looks very promising!

Page and line numbers refer to those in the clean version of the revised manuscript.

**Major Comments**
- None

**Minor Comments**
- p.1, l.4: OK, even in the tortured world of academic acronyms, I'm really struggling to see how the authors are getting AGILE out of 'Open Global Glacier Data Assimilation Framework'. Obviously, the setup can be given any name the authors want, but the way it's currently written implies it's some sort of acronym that the reader should be able to derive from the words and it...just...isn't? At least not without just picking an entirely random subset of letters. Either call it OGGDAF (don't call it OGGDAF), or write it as something like 'We present the Open Global Glacier Data Assimilation Framework, named AGILE' to make it clear that it's just a name that the authors like rather than an abbreviation of something in the long-form name. I'd be tempted to say change it in the paper title too, but I think that would make the title a bit unwieldy, so just making the change in the abstract and then in the introduction at l. 65 should suffice to unconfuse the reader.
- Figure 1: I might suggest adding, either to the caption or as a legend in the figure, something explaining that the things in green are the control variables that the model is trying to match. They're all listed in the caption, so it's not too bad to make the logical jump, but including the colour information somewhere would help the reader.
- p.6, l.132: Were they discussed in Section 1? The introduction definitely covers the context, but I don't think it specifically discusses the choice of these particular variables and their implications in reality, unless l.75-79 is meant? Which is a quite high-level summary and doesn't name the specific variables. Possibly, I'm getting confused, but maybe there needs to be some additional information in the introduction, or this reference needs to be modified.
- p.9, l.233-234: Might be worth adding where these glaciers actually are so the reader can understand better that they do represent a range of different climates (I certainly know where Aletsch and Baltoro are, I think I have a rough idea for Peyto, no idea for Artesonraju)
- Figure 3 caption: typo for Aletsch ('Altesch')
- Figure 4 caption: 'Panels a and d show…', 'Panels c and f illustrate...'
- Figure 5 caption: same idea – if you've got two panels showing something, don't put the verb in the 3rd-person singular.
- Figure 6 caption: 'The gray shaded area…', 'after 20 iterations.'
- p.18, l.408: 'where the number of model runs increases substantially…' - 'a lot' is a bit colloquial
- p.21, l.428: 'depending on if'

---

## Referee Comment (RC2)

**Review of Schmitt et al. (2025) 'The Open Global Glacier Data Assimilation Framework (AGILE) v0.1'**

The manuscript presents a new data assimilation framework named AGILE for OGGM. It is based on automatic differentiation and the optimization algorithm L-BFGS-B, and it improves glacier geometry, specifically bed height and volume along the flowline. The method is tested and analyzed using synthetic observations, namely distributed surface elevation and total volume estimates for one year, as well as geodetic mass balance over a 20-year period. The results from 12 different settings show that AGILE improves the initial guess of glacier geometry in an efficient and consistent way. Only a noisy pattern at the upper end of the flowline persists, and the Baltoro Glacier appears to be challenging for both the initial guess and AGILE. An analysis of the weighting between the cost function and regularization provides first insights into the impact of regularization; however, as the authors note, the findings are limited to idealized settings and may not be directly transferable to real observations.

The framework is well presented, and the description of the experiments is comprehensive. The use of automatic differentiation is an elegant and efficient tool for optimizing a large set of control variables. The framework has the potential to significantly improve glacier projections. The authors also made a notable effort to ensure reproducibility through the published code on GitHub and Zenodo, as well as the use of Docker. I consider the manuscript highly suitable for Geoscientific Model Development after addressing some minor structural and clarity issues in the Results section, clarifying the choice of observations, and improving the guidance for running the example code. Some of the comments are suggestions for the authors' consideration to further enhance the presentation quality.

**General comments**

- 1. The choice of total ice volume as an observation could be explained more clearly, as it is not a directly measured quantity but depends on assumptions about glacier geometry or dynamics. It would be helpful if the authors could elaborate on how such a volume estimate could realistically be obtained in practice. The results already show that AGILE can improve the initial guess even without this information, which is encouraging. Using pointwise measurements from ground-penetrating radar would be a more realistic option, even if such data are rare. Likewise, spatially distributed elevation change data could be a valuable alternative to a single geodetic mass balance value per glacier, as they allow for easier comparison along a flowline. The choice of observations could be better justified by referencing actual observational datasets.
- 2. The structure of the Results section could be improved. The separation into Sections 4.2 and 4.3 likely reflects the incremental development of the study but does not read

- smoothly. Sections 4.3.1, 4.2.1, and 4.2.3 could be grouped under "Proof of Concept for AGILE Functionality" and "Influence of First Guess." The current title "Functionality of Minimizing the Cost" is not very informative.
- 3. Sections 4.2.2 and 4.3.3 have similar scopes, with 4.2.2 focusing only on Aletsch Glacier and 4.3.3 extending the analysis to four glaciers and three dynamic states. It is reasonable to present one detailed example, but I suggest merging 4.2.2 and 4.3.3 for a smoother presentation. The results from the four glaciers in the λ-sensitivity study could also be aggregated. Although the authors note that the influence of λ varies between glaciers, similar trends can be observed, especially in the three-observation setting. The results for individual glaciers could be moved to the Supplementary Material, while a single figure aggregating the results across all eight settings would make it easier to generalize an appropriate λ value for untested glaciers and keep the manuscript uncluttered. This is a suggestion for the authors' consideration and can be omitted if an aggregated figure does not provide meaningful insights.
- 4. Although the code is provided via GitHub and Docker, I was unable to execute a simple showcase scenario e.g. reproducing Figure 5. I am not very familiar with Docker nor OGGM, so this may be due to my setup. I would suggest including a "first\_steps" Jupyter notebook, similar to the one in the GitHub-repository, directly in the Docker image. Additionally, the example for agile1d seems somewhat hidden at the end of "first\_steps.ipynb" and may not be fully compatible with the Docker image. I would appreciate a step-by-step instruction in the README to run a showcase setting.

**Specific comments**

- **Line 13:** What is meant by "diffusive nature of glacier systems"? Please provide a reference or clarify the intended meaning.
- Line 88: Is there a way to estimate the uncertainty of the control variables?
- **Line 155:** Emphasizing that *i* starts at 0 suggests that the highest point might be at *i* = 1. I recommend omitting the specific index and simply stating that an additional grid point is included.
- **Line 166:** The distinction between variables and parameters could be improved. The term *control variables* should be used consistently. The phrase "unknown parameters or variables" is followed by no variables in the example; it could instead refer to flowline-geometry variables, because these are later referred to as control variables.
- **Line 187f:** The chosen percentages seem arbitrary as well as the ten extra grid points. What happens if the initial guess is very poor?

- Line 194: How can these values be used as control variables? Please clarify.
- Line 210: A short description of the algorithm would be helpful, especially since the low number of model runs per iteration is emphasized in the results. Additionally the relation to 4D-Var should be explained, since it is mentioned in the abstract and introduction.
- Line 223: How can *volume* be considered an observation? Please explain.
- **Line 255:** A reference for these uncertainties would strengthen the statement. The choice could influence the results under the different cost function settings.
- **Figure 2:** Why do the different dynamic states have different volumes in 1980? How is the initial volume chosen? This might need to be added to Sect. 3.1.
- Line 280: The temporal mismatch mentioned here is unclear. Please clarify.
- **Figure 5:** Using a color gradient between iteration colors and increasing the step size to five could allow convergence to be visualized without relying on the legend. This is more a personal preference.
- **Figure 6:** The figure appears cramped vertically, and the y-axis is missing an additional tick (e.g., 0). If possible, give the figure more vertical space. An aggregation with Figure 9 could be considered.

**Technical corrections**

- Line 66: 4D-Var
- Line 166: optimized
- Line 178: area—height
- **Figure 2:** The -1 should be in the exponent.
- **Figure 3:** Remove the first *"first guess"* in the caption.
- **Figure 4:** Be consistent in notation, use either (a–c) or (a to c).
- Line 368: (Figure 4, panel g)
- Figures 6 & 9: Title and caption, use after 20 iterations (not after the 20. Iteration).

- Figure 6: Caption, "The gray shaded area ..."
- Line 409: minimization, minimizing
- Line 427: analyzed
- **Line 505**: *semi-implicit* (no capital letter)
- Line 534: generalized

---

## Author Comment (AC1)

Thank you to the two anonymous reviewers for their helpful and encouraging feedback. We also appreciate the time and effort the reviewers and editor have spent on our manuscript. We have addressed the comments, and our responses to specific points are provided below in blue text.

**Review by Anonymous Referee #1**

**Summary**

This paper presents a new workflow built on OGGM that allows improved inversion of glaciers and the construction of dynamically consistent states suitable for use as initialisation for prognostic simulations. The method is applied to idealised versions of four glaciers as a proof-of-concept and shown to work quickly and effectively in refining the coarse initial guess at the glacier profile derived using two different current methods and then running forward to hit the target 2020 state across all four glaciers and regardless of whether the glaciers are assumed to be advancing, retreating or in dynamic equilibrium.

This is a very thorough, well-written paper that convincingly presents the new workflow. The only thing missing is an application to real cases, but that is very deliberately set outside the scope of this paper by the authors; I only hope that they do shortly follow up with such a study! Certainly, as presented, the method is undeniably effective in the synthetic cases included in this paper. Otherwise, I have only a few comments of a very minor nature, so I recommend the paper be accepted subject to minor revisions. Congratulations to the authors – this looks very promising!

Thank you!

Page and line numbers refer to those in the clean version of the revised manuscript.

**Major Comments**
- None

**Minor Comments**
- p.1, l.4: OK, even in the tortured world of academic acronyms, I'm really struggling to see how the authors are getting AGILE out of 'Open Global Glacier Data Assimilation Framework'. Obviously, the setup can be given any name the authors want, but the way it's currently written implies it's some sort of acronym that the reader should be able to derive from the words and it...just...isn't? At least not without just picking an entirely random subset of letters. Either call it OGGDAF (don't call it OGGDAF), or write it as something like 'We present the Open Global Glacier Data Assimilation Framework, named AGILE' to make it clear that it's just a name that the authors like rather than an abbreviation of something in the long-form name. I'd be tempted to say change it in the paper title too, but I think that would make the title a bit unwieldy, so just making the change in the abstract and then in the introduction at l. 65 should suffice to unconfuse the reader.

Thank you for pointing this out. We adapted the title to reflect that this is indeed not a true acronym : "**AGILE v0.1: The Open Global Glacier Data Assimilation Framework**".
Further, we changed the wording as suggested in the abstract and in the introduction, by using 'named AGILE'.

- Figure 1: I might suggest adding, either to the caption or as a legend in the figure, something explaining that the things in green are the control variables that the model is trying to match. They're all listed in the caption, so it's not too bad to make the logical jump, but including the colour information somewhere would help the reader.

  Thank you for the suggestion. We have added the color information to the caption as:

  "The mismatch between the model output and observations **(depicted in green)**,..."

  Just to clarify wording, the observations shown in the figure are not the same as the control variables. The control variables refer to the model parameters that are iteratively adjusted to minimize the cost function and reduce the mismatch with the observations. These control variables are not directly shown in the figure. To avoid such confusions in the future we adapted a sentence in the caption:

  "If the cost function has not reached a minimum, all control variables **(i.e., the unknown variables we aim to estimate)** are updated simultaneously …"

- p.6, l.132: Were they discussed in Section 1? The introduction definitely covers the context, but I don't think it specifically discusses the choice of these particular variables and their implications in reality, unless l.75-79 is meant? Which is a quite high-level summary and doesn't name the specific variables. Possibly, I'm getting confused, but maybe there needs to be some additional information in the introduction, or this reference needs to be modified.

  Those are listed and discussed in L24 - L36, but we have not explicitly linked this. With implications for real world application we meant the discussion about shifting towards glacier-specific observations as datasets have become available (L25-L29). To make this connection to Section 1 more clear we adapted the sentence at L132 as following:

  "The choice of these variables and the implications for real-world applications **were** discussed in Sect. 1**: these are the currently available global datasets used by the glacier models participating in GlacierMIP3."**

- p.9, l.233-234: Might be worth adding where these glaciers actually are so the reader can understand better that they do represent a range of different climates (I certainly know where Aletsch and Baltoro are, I think I have a rough idea for Peyto, no idea for Artesonraju)

  We agree that this additional information helps readers better understand the climatic diversity of the selected glaciers. We have added the mountain range and continent

for each glacier in parentheses: "We selected the glaciers Aletsch **in the Alps (Europe)**, Artesonraju **in the Cordillera Blanca (South America)**, Baltoro **in the Karakoram (Asia)**, and Peyto **in the Canadian Rockies (North America)** because they represent a range of different climates and glacier area sizes."

- Figure 3 caption: typo for Aletsch ('Altesch')

  Thank you, corrected .

- Figure 4 caption: 'Panels a and d show…', 'Panels c and f illustrate...'

  corrected

- Figure 5 caption: same idea – if you've got two panels showing something, don't put the verb in the 3rd-person singular.

  corrected

- Figure 6 caption: 'The gray shaded area…', 'after 20 iterations.'

  corrected

- p.18, l.408: 'where the number of model runs increases substantially…' - 'a lot' is a bit colloquial

  corrected

- p.21, l.428: 'depending on if'

  corrected

---

## Author Comment (AC2)

**Review by Anonymous Referee #2**

The manuscript presents a new data assimilation framework named AGILE for OGGM. It is based on automatic differentiation and the optimization algorithm L-BFGS-B, and it improves glacier geometry, specifically bed height and volume along the flowline. The method is tested and analyzed using synthetic observations, namely distributed surface elevation and total volume estimates for one year, as well as geodetic mass balance over a 20-year period. The results from 12 different settings show that AGILE improves the initial guess of glacier geometry in an efficient and consistent way. Only a noisy pattern at the upper end of the flowline persists, and the Baltoro Glacier appears to be challenging for both the initial guess and AGILE. An analysis of the weighting between the cost function and regularization provides first insights into the impact of regularization; however, as the authors note, the findings are limited to idealized settings and may not be directly transferable to real observations.

The framework is well presented, and the description of the experiments is comprehensive. The use of automatic differentiation is an elegant and efficient tool for optimizing a large set of control variables. The framework has the potential to significantly improve glacier projections. The authors also made a notable effort to ensure reproducibility through the published code on GitHub and Zenodo, as well as the use of Docker. I consider the manuscript highly suitable for Geoscientific Model Development after addressing some minor structural and clarity issues in the Results section, clarifying the choice of observations, and improving the guidance for running the example code. Some of the comments are suggestions for the authors' consideration to further enhance the presentation quality.

Thank you!

**General comments**

1. The choice of total ice volume as an observation could be explained more clearly, as it is not a directly measured quantity but depends on assumptions about glacier geometry or dynamics. It would be helpful if the authors could elaborate on how such a volume estimate could realistically be obtained in practice. The results already show that AGILE can improve the initial guess even without this information, which is encouraging. Using pointwise measurements from ground-penetrating radar would be a more realistic option, even if such data are rare. Likewise, spatially distributed elevation change data could be a valuable alternative to a single geodetic mass balance value per glacier, as they allow for easier comparison along a flowline. The choice of observations could be better justified by referencing actual observational datasets.

   Thank you for the question and the suggestions for additional observations.

   The choice to use ice volume as an observation, even though it is not directly measured, reflects the common practice in large-scale glacier modelling (e.g. in GlacierMIP3 all models are initialised with a reference ice thickness product). These

models often treat total glacier volume as if it were an observed quantity, since it is a key property that needs to be constrained for a plausible initialization. Estimating total glacier volume without observational constraints remains an active area of research and is beyond the scope of this study.

We have added a paragraph to the introduction (line 37) to explain the role of total volume in large-scale modelling, and adapted the beginning of the following paragraph:

"**Although widely used, the consensus ice thickness estimate from Farinotti et al. (2019) is not a direct observation but rather the mean result from an ensemble of models. In large-scale glacier modelling, however, it is often treated as an observed quantity (i.e. a target to match via calibration) because glacier volume is essential for estimating a plausible initial glacier state. Improving glacier-specific volume estimates in absence of observations remains an active area of research, with several methods currently being developed and tested in selected regions, aiming for future global applications (e.g., Cook et al., 2023; van Pelt and Frank, 2025).**

**To better approximate initial conditions and account for glacier evolution prior to the outline date, three GlacierMIP3 models include a dynamic spin-up during initialization.** In these cases, a past glacier state is defined, and the model is run forward in time to match specific targets. …
"

However, the reviewer is right to assume that AGILE's potential for estimating total ice volume with observational constraints (e.g. point thicknesses)  could be  part of a real-world application of the system.

In this study, however, we focused exclusively on datasets that are currently available globally for all glaciers to ensure broad applicability. Future studies should certainly explore the integration of other observation types. Doing so could not only improve model performance but also help identify which types of observations would be most valuable for future modelling efforts, potentially informing the design of future data collection strategies.

Actual observation datasets are listed in lines 24–36. To clarify the connection to our choice of variables, we revised the sentence at line 132 to:

"The choice of these variables and the implications for real-world applications **were** discussed in Sect. 1**: these are the currently available global datasets used by the glacier models participating in GlacierMIP3.**"

2. The structure of the Results section could be improved. The separation into Sections 4.2 and 4.3 likely reflects the incremental development of the study but does not read smoothly. Sections 4.3.1, 4.2.1, and 4.2.3 could be grouped under "Proof of Concept

for AGILE Functionality" and "Influence of First Guess." The current title "Functionality of Minimizing the Cost" is not very informative.

3. Sections 4.2.2 and 4.3.3 have similar scopes, with 4.2.2 focusing only on Aletsch Glacier and 4.3.3 extending the analysis to four glaciers and three dynamic states. It is reasonable to present one detailed example, but I suggest merging 4.2.2 and 4.3.3 for a smoother presentation. The results from the four glaciers in the λ-sensitivity study could also be aggregated.

We thank the reviewer for the helpful suggestions regarding the merging of paragraphs. We agree that this restructuring improves the clarity of the results presentation and avoids redundancy by preventing similar results from being discussed in different parts of the manuscript.

Following the reviewer's advice, we reorganized Sections 4.2 and 4.3 and merged selected subsections accordingly. The original text was only slightly modified to ensure smooth transitions between the newly merged sections. We also updated the figure layout of Figures 4 and 7 to match the revised structure and merged Figures 6 and 9, as recommended in a comment below.

The updated structure and the origin of the content (shown in square brackets) are as follows:

4.2 Proof of concept for AGILE functionality *[merging 4.2.1, parts of 4.3.1 and 4.3.2]*
    4.2.1 Aletsch retreating
    4.2.2 Performance across glacier geometries and dynamic states
    4.2.3 Distributed differences along the flowline
4.3 Influence of first guess *[merging 4.2.3 and parts of 4.3.1]*
4.4 Different cost function settings *[merging 4.2.2 and 4.3.3]*

Although the authors note that the influence of λ varies between glaciers, similar trends can be observed, especially in the three-observation setting. The results for individual glaciers could be moved to the Supplementary Material, while a single figure aggregating the results across all eight settings would make it easier to generalize an appropriate λ value for untested glaciers and keep the manuscript uncluttered. This is a suggestion for the authors' consideration and can be omitted if an aggregated figure does not provide meaningful insights.

Thank you for the suggestion to create an aggregated figure. While we appreciate the idea, we do not believe such a figure would provide meaningful additional insights within the context of this study. This is due to the strong assumptions underlying the creation of our synthetic test cases. As a result, any aggregated result would only be valid for selecting an appropriate λ value for untested glaciers within this idealized setting, and cannot be directly transferred to real-world applications.

However, we agree that a similar analysis could be valuable in future studies focused on real glaciers with more validation data. In that context, it may become possible to generalize findings and assess their applicability to untested glaciers.

4. Although the code is provided via GitHub and Docker, I was unable to execute a simple showcase scenario e.g. reproducing Figure 5. I am not very familiar with Docker nor OGGM, so this may be due to my setup. I would suggest including a "first_steps" Jupyter notebook, similar to the one in the GitHub-repository, directly in the Docker image. Additionally, the example for agile1d seems somewhat hidden at the end of "first_steps.ipynb" and may not be fully compatible with the Docker image. I would appreciate a step-by-step instruction in the README to run a showcase setting.

Thank you for pointing this out. To ensure reproducibility, I created a bash script that downloads the GitHub repository and the corresponding Docker image, and runs the Aletsch retreating case as an example. Additionally, I provided a second script that uses the experiment data to recreate Figure 5 for the Aletsch retreating case. Instructions for using these scripts have been added to the README of the repository, including guidance for users who wish to rerun all experiments, not just Aletsch retreating. These changes have also been uploaded to Zenodo, and the updated link has been included in the manuscript.

**Specific comments**

● Line 13: What is meant by "diffusive nature of glacier systems"? Please provide a reference or clarify the intended meaning.

Thank you for pointing out this ambiguity. We have revised the sentence to clarify that glacier dynamics can be described by a diffusion equation, which inherently leads to a loss of information over time. The updated sentence now reads:

"We also examine the potential to reconstruct earlier glacier states (e.g., in 1980) without direct observations and find that this is fundamentally limited **because** glacier dynamics **are governed by a diffusion equation, which leads to a loss of information about past states over time**, even in an idealized setting."

● Line 88: Is there a way to estimate the uncertainty of the control variables?

Thank you for the question. This is indeed a very relevant aspect for real-world applications, and we have not yet explored these possibilities within the AGILE workflow. We plan to investigate this further in an upcoming study focused on real-world glaciers, where additional validation data will be available. Potential approaches could include Bayesian methods (e.g., Rounce et al., 2020) or Monte Carlo simulations (e.g., Machguth et al., 2008). However, both techniques require generating an ensemble of results, which may pose computational challenges for global-scale applications of inverse workflows, even if AGILE is relatively cheap in its flowline version. If this proves to be a limiting factor, a possible alternative could be a method similar to that discussed in Recinos et al. (2023), which avoids ensemble generation but requires a more advanced automatic differentiation infrastructure than AGILE currently supports.

- Line 155: Emphasizing that i starts at 0 suggests that the highest point might be at i = 1. I recommend omitting the specific index and simply stating that an additional grid point is included.

  Thank you for the suggestion. We omitted the specific index and adapted the sentence as suggested.

- Line 166: The distinction between variables and parameters could be improved. The term control variables should be used consistently. The phrase "unknown parameters or variables" is followed by no variables in the example; it could instead refer to flowline-geometry variables, because these are later referred to as control variables.

  Thank you for pointing this out. Instead of 'parameters or variables' we decided to stick to use only 'variables' when talking about control variables. We adapted this throughout the manuscript.

- Line 187f: The chosen percentages seem arbitrary as well as the ten extra grid points. What happens if the initial guess is very poor?

  Thank you for the comment. We agree that the selection of the control variable bounds is somewhat arbitrary and could be too restrictive if the initial guess is very poor. For real-world applications, these bounds need to be carefully assessed. Ideally, they should be as narrow as possible to incorporate prior knowledge, while still being wide enough to allow AGILE the flexibility to explore a sufficient range of values.

  In our synthetic test cases, we used relatively wide bounds (±60% for the glacier bed height and ±40% for the initial volume at each grid point), reflecting low confidence in the initial guess. For real-world applications, a sensitivity analysis is recommended to evaluate the effect of different bounds. However, in our idealized setup, such an analysis would not provide additional insights, since the goal is to demonstrate the functionality of AGILE under controlled conditions. In general, the less confidence one has in the initial guess, the broader the bounds should be.

  Regarding the ten additional grid points: this number could potentially be estimated by extrapolating from the observed outline and assessing how much larger the glacier area was in the initialization year (1980 in our case). If this estimate is too small, AGILE may fail to reconstruct the correct glacier extent for 1980. These missing grid points are located at the glacier terminus and, for retreating glaciers (which are particularly relevant given the globally observed glacier retreat, e.g. The GlaMBIE Team, 2025) they will melt first and have only a minimal influence on today's glacier state.

  Moreover, as demonstrated by our results, reconstructing past glacier states without direct observations is fundamentally limited and not the primary objective of our setup. Instead, the goal is to initialize a glacier model with a dynamically consistent present-day state by using available past observations.

- Line 194: How can these values be used as control variables? Please clarify.

  Thank you for pointing out this ambiguity. Internally, the control variables are divided by the initial surface width at each grid point to retrieve the glacier bed heights, which are used in the dynamic model run. To clarify this, we have added the following sentence after the one in Line 194:

  "To account for this, we multiply the glacier bed height by the initial surface width at each grid point and use the resulting values as our control variables. **During each iteration, these control variables are divided by the same initial surface width to recover the estimated glacier bed, which is then used in the dynamic simulation."**

- Line 210: A short description of the algorithm would be helpful, especially since the low number of model runs per iteration is emphasized in the results.

  We added a short description of the algorithm and an additional paragraph explaining the relation to 4D-Var. The two paragraphs now reads as:

  **"**The minimization process is conducted using the L-BFGS-B algorithm, as implemented in the Python package SciPy (Virtanen et al., 2020, https://www.scipy.org/). **L-BFGS-B is a limited-memory quasi-Newton optimization algorithm that, given a smooth objective function, its gradient, an initial guess, and simple bound constraints on the control variables, iteratively approximates the inverse Hessian to efficiently find a constrained minimum (Byrd et al., 1995; Zhu et al., 1997; Morales and Nocedal, 2011). Similar algorithms have been used in comparable geophysical studies (e.g., Goldberg and Heimbach, 2013; Fürst et al., 2017). In our case,** we set the bounds for the control variables to the minimum and maximum values defined in Sect. \ref{sec:control_variabls}.

  Additionally the relation to 4D-Var should be explained, since it is mentioned in the abstract and introduction.

  Thank you for pointing out the missing definition of 4D-Var and the need to clarify which concepts we are adopting. To address this, we have added the following clarification to the introduction at L66:

  "To move closer to the transient calibration of large scale glacier models, we present a proof-of-concept for the Open Global Glacier Data Assimilation Framework, named AGILE. AGILE is based on a time-dependent variational assimilation approach, inspired by 4D-Var methods (Lorenc, 1997)**, which extend the spatial domain (x, y, z) of 3D-Var or snapshot approaches by adding time as a fourth dimension. Following this concept, AGILE treats time as a coordinate during assimilation, enabling the consistent integration of temporally distributed observations.**

  **In line with this concept, AGILE** iteratively adjusts all control variables …"

- Line 223: How can volume be considered an observation? Please explain.

  We added a paragraph, as described in the answer to the general comment 1.

- Line 255: A reference for these uncertainties would strengthen the statement. The choice could influence the results under the different cost function settings.

  Yes, you are right that the choice of uncertainties influences the results when using different cost function settings, as these uncertainties serve to scale the contributions of different types of observations relative to each other. The specific values used in our idealized experiments are chosen to reflect the typical order of magnitude of uncertainties reported in available datasets. We have added this clarification to the manuscript as follows:

  "We define the associated measurement uncertainties **to reflect the typical order of magnitude found in reported real-world data, setting them** as $\sigma_{sfc}$ = 10 m **(Uuemaa et al., 2020)**, $\sigma_{\Delta M}$ = 100 kg m$^{-2}$ yr$^{-1}$ **(Hugonnet et al., 2021)** and $\sigma_V$ = 10% of $V_{2000}$ **(Farinotti et al., 2019)**."

- Figure 2: Why do the different dynamic states have different volumes in 1980? How is the initial volume chosen? This might need to be added to Sect. 3.1.

  Thank you for pointing at this missing information. We added the following paragraph to Sect 3.1 L238:

  **"For the advancing and retreating cases, we first performed a 60-year dynamic simulation using climate data from 1920 to 1980, followed by a second simulation from 1980 to 2020. For each glacier, dynamic state, and simulation period, temperature biases were applied to ensure that the total volume changes from 1980 to 2020 were of similar magnitude for both retreating and advancing conditions. For the equilibrium state, we defined a constant mass balance profile based on the average over specific historical periods (Aletsch: 1988–1998, Arensonraju: 1981–1983, Baltoro: 1981–1993, Peyto: 1948–1966). This profile was applied for 120 years before 1980 to allow the glacier to reach equilibrium, and the same mass balance was then used from 1980 to 2020.** Figure \ref{fig:aletsch_geometry_creation} panels c and d show the resulting mass balance and volume evolution for the different dynamic states of the Aletsch glacier. For the other glaciers, see Figure S1 panels c and d for Artesonraju, panels g and h for Baltoro, and panels k and l for Peyto**."**

- Line 280: The temporal mismatch mentioned here is unclear. Please clarify.

  With 'temporal mismatch', we emphasize again that our first guess for the 1980 volume is based on a volume estimate from the year 2000. This mismatch is explained in more detail in Section 2.5. To clarify this point, we have revised the sentence as follows:

"At this stage, we focus only on the glacier bed, **as we acknowledge the temporal mismatch introduced by using a volume distribution estimate from 2000 as the first guess for the 1980 state** (see Sect. 2.5)."

- Figure 5: Using a color gradient between iteration colors and increasing the step size to five could allow convergence to be visualized without relying on the legend. This is more a personal preference.

Thank you for the suggestion. We adapted the figure to use a color gradient and changed the step size to five. See below the updated figure.

[Figure]

- Figure 6: The figure appears cramped vertically, and the y-axis is missing an additional tick (e.g., 0). If possible, give the figure more vertical space. An aggregation with Figure 9 could be considered.

Thank you for the suggestion. We have combined Figures 6 and 9 into a single aggregated figure, increased the vertical size, and added a tick at 0 on the y-axis, as suggested. The updated figure is shown below:

[Figure]

**Technical corrections**

- Line 66: 4D-Var  adapted
- Line 166: optimized  adapted
- Line 178: area–height  adapted
- Figure 2: The –1 should be in the exponent.  adapted, also in the supplement figure
- Figure 3: Remove the first "first guess" in the caption.  adapted
- Figure 4: Be consistent in notation, use either (a–c) or (a to c).  adapted in Fig4, Fig5 and Fig9
- Line 368: (Figure 4, panel g)  adapted
- Figures 6 & 9: Title and caption, use after 20 iterations (not after the 20. Iteration).  Merged the two figures and adapted the title and caption as suggested

- Figure 6: Caption, "The gray shaded area …"  adapted
- Line 409: minimization, minimizing  adapted
- Line 427: analyzed  adapted
- Line 505: semi-implicit (no capital letter)  adapted
- Line 534: generalized  adapted

References:

Cook, S. J., Jouvet, G., Millan, R., Rabatel, A., Zekollari, H., and Dussaillant, I.: Committed Ice Loss in the European Alps Until 2050 Using a Deep‑Learning‑Aided 3D Ice‑Flow Model With Data Assimilation, Geophysical Research Letters, 50, https://doi.org/10.1029/2023gl105029, 2023.

Farinotti, D., Huss, M., Fürst, J. J., Landmann, J., Machguth, H., Maussion, F., and Pandit, A.: A consensus estimate for the ice thickness distribution of all glaciers on Earth, Nat. Geosci., 12, 168–173, https://doi.org/10.1038/s41561-019-0300-3, 2019.

Frank, T. and van Pelt, W. J. J.: Ice volume and thickness of all Scandinavian glaciers and ice caps, J. Glaciol., 70, https://doi.org/10.1017/jog.2024.25, 2024.

The GlaMBIE Team: Community estimate of global glacier mass changes from 2000 to 2023, Nature, 639, 382–388, https://doi.org/10.1038/s41586-024-08545-z, 2025.

Hugonnet, R., McNabb, R., Berthier, E., Menounos, B., Nuth, C., Girod, L., Farinotti, D., Huss, M., Dussaillant, I., Brun, F., and Kääb, A.: Accelerated global glacier mass loss in the early twenty-first century, Nature, 592, 726–731, https://doi.org/10.1038/s41586-021-03436-z, 2021.

Lorenc, A. C.: Development of an Operational Variational Assimilation Scheme (gtSpecial IssueItData Assimilation in Meteology and Oceanography: Theory and Practice), Journal of the Meteorological Society of Japan. Ser. II, 75, 339–346, https://doi.org/10.2151/jmsj1965.75.1b_339, 1997

Machguth, H., Purves, R. S., Oerlemans, J., Hoelzle, M., and Paul, F.: Exploring uncertainty in glacier mass balance modelling with Monte Carlo simulation, The Cryosphere, 2, 191–204, https://doi.org/10.5194/tc-2-191-2008, 2008.

Recinos, B., Goldberg, D., Maddison, J. R., and Todd, J.: A framework for time-dependent ice sheet uncertainty quantification, applied to three West Antarctic ice streams, The Cryosphere, 17, 4241–4266, https://doi.org/10.5194/tc-17-4241-2023, 2023.

Rounce, D. R., Khurana, T., Short, M. B., Hock, R., Shean, D. E., and Brinkerhoff, D. J.: Quantifying parameter uncertainty in a large-scale glacier evolution model using Bayesian inference: application to High Mountain Asia, J. Glaciol., 66, 175–187, https://doi.org/10.1017/jog.2019.91, 2020.

Uuemaa, E., Ahi, S., Montibeller, B., Muru, M., and Kmoch, A.: Vertical Accuracy of Freely Available Global Digital Elevation Models (ASTER, AW3D30, MERIT, TanDEM-X, SRTM, and NASADEM), Remote Sensing, 12, 3482, https://doi.org/10.3390/rs12213482, 2020.